

# Temporal stability of a new 40-year daily AVHRR Land Surface Temperature dataset for the Pan-Arctic region

Sonia Dupuis [1], Frank-Michael Göttsche [2], and Stefan Wunderle [1]

[1]Institute of Geography and Oeschger Center for Climate Change Research, University of Bern, Switzerland
[2]Institute of Meteorology and Climatology Research, Karlsruhe Institute of Technology, Germany

**Correspondence:** Sonia Dupuis  (sonia.dupuis@unibe.ch)

**Abstract.** Land surface temperature (LST) gained increased attention in cryospheric research. While various global satellite LST products are available, none of them is specially designed for the Pan-Arctic region. Based on the recently published EUMETSAT Advanced Very High Resolution Radiometer (AVHRR) fundamental data record (FDR), a new LST product (1981-2021) with daily resolution is developed for the Pan-Arctic region. Validation shows good accuracy with an average

mean absolute error (MAE) of 1.71 K and a MAE range of 0.62–3.07 K against in situ LST data from the Surface Radiation Budget Network (SURFRAD) and Karlsruhe Institute of Technology (KIT) sites. Long-term stability, a strong requirement for trend analysis, is assessed by comparing LST with air temperatures from ERA5-Land (T2M) and air temperature data from the EUSTACE (www.eustaceproject.org) global station dataset. Long-term stability might not be fulfilled mainly due to the orbit drift of the NOAA satellites. Therefore, the analysis is split into two periods: the arctic winter months, which are unaffected by

solar illumination and, therefore, orbital drift, and the summer months. The analysis for the winter months results in correlation values (r) of 0.44-0.83, whereas for the summer months (r) ranges between 0.37-0.84. Analysis of anomaly differences revealed instabilities for the summer months at a few stations. The same stability analysis for the winter months revealed only one station with instabilities in comparison to station air temperature. Discrepancies between the temperature anomalies recorded at the stations and ERA5-Land T2M were also found. This highlights the limited influence of orbital drift on the LST product, with

the winter months presenting good stability across all stations, which makes these data a valuable source for studying LST changes in the Pan-Arctic region over the last 40 years. This study concludes by presenting LST trend maps (1981-2021) for the entire region, revealing distinct warming and cooling patterns.

## 1  Introduction

In recent decades, the warming in the Arctic has been much faster than in the rest of the world. Studies (Chylek et al., 2022; Rantanen et al., 2022) indicate a warming up to four times faster since 1979. This phenomenon, known as Arctic amplification, is visible in both instrumental records and model simulation (Dada et al., 2022). The Arctic Monitoring & Assessment Pro-



gramme (AMAP) highlights diminishing long cold spells and increasing high extreme temperatures, leading to rapid changes in the cryosphere (AMAP, 2021). These episodes affect the sensitive arctic ecosystem, vegetation dynamics, large-scale circu-

lation patterns and the distribution of snow, ice and permafrost (Maturilli et al., 2019). Permafrost, a crucial component of arctic ecosystems, is particularly sensitive to increasing air temperatures and changes in the snow regime. Thawing permafrost affects the stability of the bedrock, damages infrastructures, and releases massive quantities of organic carbon (Christensen et al., 2004; Miner et al., 2022). These potential threats highlight the importance of monitoring climate variables such as temperature in the arctic regions (Hachem et al., 2012; Urban et al., 2013). Rapid changes in land surface temperature (LST) patterns have

been observed in the Arctic region (Reiners et al., 2021). LST can be used as an indicator of the thermal state of the ground and has, in the last decade, been increasingly used in arctic research and permafrost modelling (Westermann et al., 2009; Obu et al., 2019; Batbaatar et al., 2020; Nielsen-Englyst et al., 2021). LST observations are available from in situ stations or climate models. However, these sources are insufficient to spatially resolve land surface characteristics and their response to climate change at a hemispheric scale (Nitze et al., 2018; Bartsch et al., 2023).

In contrast, satellite data can derive spatially comprehensive information on LST dynamics (Li et al., 2013, 2023b). LST is mainly derived from thermal infrared (TIR) radiation measured by satellites with algorithms based on radiative transfer (RT) equations (Li et al., 2013). Passive microwave (MW) are another source for LST data. MW measurements are less affected by clouds than TIR data, but emissivity is challenging to derive for these wavelengths, especially over snow-covered ground (Jiménez et al., 2017; Ermida et al., 2017). LST is a critical parameter in Earth's surface and water energy balance and is widely

exploited across different research fields: cryosphere, geology, vegetation monitoring, hydrology, and urban management (Li et al., 2013; Guillevic et al., 2018). From a climate perspective, LST is needed to evaluate land surface and land-atmosphere exchange processes, constrain surface energy budgets and model parameters, and provide observations of surface temperature change globally and in key regions (Guillevic et al., 2018). LST is defined as an essential climate variable (ECV) by the the Global Climate Observing System (GCOS). To retrieve statistically significant changes in ECVs, a time series of at least 30

years is needed (WMO, 2010). Typical LST products include the Pathfinder, GLASS, MODIS, ASTER, and Landsat products (Good et al., 2022; Reiners et al., 2023; Li et al., 2023a), as well as the Spinning Enhanced Visible and Infrared Imager (SEVIRI) LST product produced within the framework of the European Organisation for the Exploitation of Meteorological Satellites (EUMETSAT) Satellite Application Facility on Land Surface Analysis (LSA-SAF) (Freitas et al., 2010; Trigo et al., 2011).

Although many satellite LST datasets with different temporal and spatial resolutions exist, only the Advanced Very High Resolution Radiometer (AVHRR) onboard the NOAA and MetOp satellites series covers over four decades. EUMETSAT published a new AVHRR fundamental data record (FDR) in May 2023 (https://navigator.eumetsat.int/product/EO:EUM:DAT: 0862). This dataset was homogeneously produced and consists of reflectance and brightness temperatures covering 1978 – 2021. It is based on reprocessed Near Real Time (NRT) observations from 17 AVHRR instruments on board NOAA satellites

TIROS-N to NOAA-19, as well as EUMETSAT satellites MetOp-A, -B, and -C (EUMETSAT, 2023a). The dataset is provided in the Global Area Coverage (GAC) resolution. Quality control of AVHRR GAC radiances and updates in the retrieval methods





offer more accurate results and better uncertainty estimates (Karlsson et al., 2023b). Notably, CLARA-A3, the third edition of the existing cloud albedo and radiation (CLARA) data record, has been produced from this FDR (Karlsson et al., 2023a).

Previous LST datasets exist that have a global coverage (Ma et al., 2020; Li et al., 2023b) or were developed for continental
(Reiners et al., 2021) or local usage. However, none of them were specifically derived for the Pan-Arctic region. Furthermore, the performance of the LST algorithms is strongly tied to the sampling of atmospheric profiles and surface properties used to calibrate the RT models. Different methodologies and data sources have been used in the past, but most LST products rely (entirely or partially) on the SeeBor database, which is built from the Thermodynamic Initial Guess Retrieval database (TIGR)-3, ERA-40 and radiosondes datasets (Borbas et al., 2005). Taking advantage of the most recent ECMWF version-5
reanalysis (ERA5), Ermida and Trigo (2022) developed a new clear-sky database for the development of LST algorithms. The synthetic database is constructed from ERA5 data chosen with a dissimilarity criterion to ensure a uniform distribution of atmospheric conditions. The clear-sky database shows a significantly wider range of conditions, and thus a wider range of brightness temperatures, than in the SeeBor database specifically. The Pan-Arctic region shows a wide range of temperatures and conditions that are not necessarily common, so it is particularly important to base the RT modelling on a robust and
representative database for that region. A new daily LST dataset, presented here for the northern high latitudes (> 50° N), is produced based on the EUMETSAT FDR and the clear-sky database (Ermida and Trigo, 2022).

The new LST dataset represents a valuable source for studying LST dynamics and its impacts on regional climates surface energy balance (Hall et al., 2012; Key et al., 2016), vegetation phenology (Li et al., 2021), temperature hot spots (Mildrexler et al., 2018) and characterizing land use/land cover dynamics. However, to perform climatological analyses, it is crucial that
the satellite LST observations are stable and robust (Waring et al., 2023).

The objectives of this paper are first the description of the LST retrieval and validation methods and second, to assess the stability of the new Pan-Arctic LST dataset. Relationships and trends with respect to the following datasets are compared: (i) in situ air temperature (Tair) measurements provided by the EUSTACE database, (ii) two-meter air temperature (T2M) data from ERA5-Land for the overlapping period (1981-2020). The comparisons are made for selected sites in the Pan-Arctic region. The
presence of trends in T2M is well established and is considered one of the major indicators of anthropogenic climate change (IPCC, 2021). Previous studies (Mildrexler et al., 2011; Hachem et al., 2012; Urban et al., 2013; Good et al., 2022) obtained a good correlation between LST and T2M/Tair, although these parameters have different physical meanings and are measured or modelled with different procedures. Hachem et al. (2012) found that LST derived from MODIS and daily near-surface air temperatures are comparable. Good (2016) noted that LST and T2M are very similar when solar heating is low or absent. The
NOAA satellites don't have a stable orbit (Ignatov et al., 2004; Latifovic et al., 2012), meaning that over the course of their operating years, their equator crossing time is shifting (Price, 1984). In the case of LST, drifting orbits could lead to artificial trends in long-time records if only one platform is considered. In the present case, the LST from the different platforms is combined (morning and afternoon overpasses), and the final product is generated over multiple satellites. However, this also means that observed trends are more complex to interpret (Lieberherr and Wunderle, 2018). Orbital drift has a more substantial
impact in the southern hemisphere and on bare soil (Sobrino et al., 2002; Gleason et al., 2002) than in the northern hemisphere.



For example, this effect has been neglected in previous studies focusing on the Arctic region (Urban et al., 2013) or lakes (Riffler et al., 2015).

In this study, the analysis is carried out for two cases: (a) polar winter, defined here as December and January, and (b) polar summer, defined here as June and July. Incident solar radiation is zero during polar winter (Lund et al., 2017; Wang and Zeng, 2014); therefore, it is expected that a trend analysis for case (a) should not be affected by orbital drift. This paper is structured as follows: The data used to produce the LST are presented in Sect. 2, and the methodology is described in Sect. 3. Validation results for the LST product are presented in 4.1. Comparisons with air temperature datasets and trend analysis are presented in 4.2, 4.3 and 4.4. Finally, Discussions and Conclusions are presented in Setc. 5 and 6.

## 2   Data

This study uses the newly generated EUMETSAT AVHRR FDR satellite dataset, one reanalysis dataset and several weather station datasets. Snow cover information is based on the snow water equivalent (SWE) dataset and snow cover fraction (SCF) from the ESA CCI+ Snow project.

### 2.1   EUMETSAT AVHRR FDR

The FDR contains AVHRR reflectance and brightness temperatures for each available orbit and channel. The daily AVHRR data from one satellite provides nearly complete coverage of the globe. AVHRR GAC measurements have been processed using the PyGAC software –a Python software package to read and transform AVHRR data in GAC format- (https://pygac. readthedocs.io/en/latest/#), including the conversion from counts to reflectance or brightness temperature and cross-calibration of the visible channels of the AVHRR sensor. The two thermal channels are calibrated following the Platinum Resistance Thermometer (PRT), space, and Internal Calibration Target (ICT) counts procedure (Kidwell, 1995; Walton et al., 1998). Detailed information is available in the PyGAC FDR ATBD (EUMETSAT, 2023b). The data are accompanied by additional metadata (such as orbit overlap and equator crossing time) and basic quality indicators (EUMETSAT, 2023c). Only satellites carrying the newer versions (AVHRR/2 and AVHRR/3) of the AVHRR are considered in this study. The second version (AVHRR/2) has five spectral channels, and the third version (AVHRR/3) has six but transmits only the data from five channels. Brightness temperature channel four is centred at 10.8 μm and brightness temperature channel five at 12 μm. The IR calibration procedure is satellite-specific, with no cross-calibration between satellites for IR channels (EUMETSAT, 2023d).

### 2.1.1   Cloud mask

Cloud cover information is obtained from the CM SAF CLARA-A3 dataset ((Karlsson et al., 2023b), which is also based on EUMETSAT AVHRR FDR (EUMETSAT, 2023). The probabilistic cloud mask (CMAPROB), included in the level-2b product, and quality flags are used. Cloud probabilities range from 0 to 100 % (EUMETSAT Satellite Application Facility on Climate Monitoring, 2023).



### 2.1.2 Snow cover data

Snow cover fraction (SCF) from optical satellite data and snow water equivalent (SWE) products from passive microwave satellite data from the ESA CCI snow project (https://climate.esa.int/en/projects/snow/) are used to obtain information on snow extent (Luojus et al., 2022). The SCF 'viewable snow' (SCFV) product is derived from the EUMETSAT AVHRR FDR and

applied for this study. A value equal to zero means that the pixel is snow-free, and 100 means that the pixel is fully covered by snow. The SWE variable is indicated in mm. Both snow products are combined to get a snow mask independent of the availability of the visible channels during polar night.

### 2.2 ERA5-Land 2m-air temperature

ERA5-Land (Muñoz Sabater et al., 2021), provided by the European Centre for Medium-Range Weather Forecasts (ECMWF),

is a downscaled version of the land component of global ERA5 reanalysis. Compared to ERA5, ERA5-Land shows better stability but reduced accuracy (Urraca and Gobron, 2023). However, the accuracy suffices to capture inter-annual variations (Rantanen et al., 2023). The ERA5-Land air temperature 2 m above the surface is compared to the LST data set.

### 2.3 In situ 2m-air temperature

In situ air temperature observations from the EU Surface Temperature for All Corners of Earth (EUSTACE) land station data

set (Brugnara et al., 2019; Rantanen et al., 2023) are used for comparison. This database stores daily minimum (Tmin) and maximum (Tmax) temperature values recorded at weather stations ∼2 m above the surface. The station data set has undergone quality controls, was homogenised and covers the period from 1850 until 2015. Weather stations from the EUSTACE database were selected according to the following criteria:

– The station lies above 50 degrees latitude;

– The underlying ground is composed of permafrost;

– Different latitudes are represented;

– The surrounding area at a station, corresponding to at least one GAC pixel, must be homogeneous;

– The time series should cover at least 30 years.

Based on these criteria, 12 stations have been selected (Fig. 1 and Table 1)





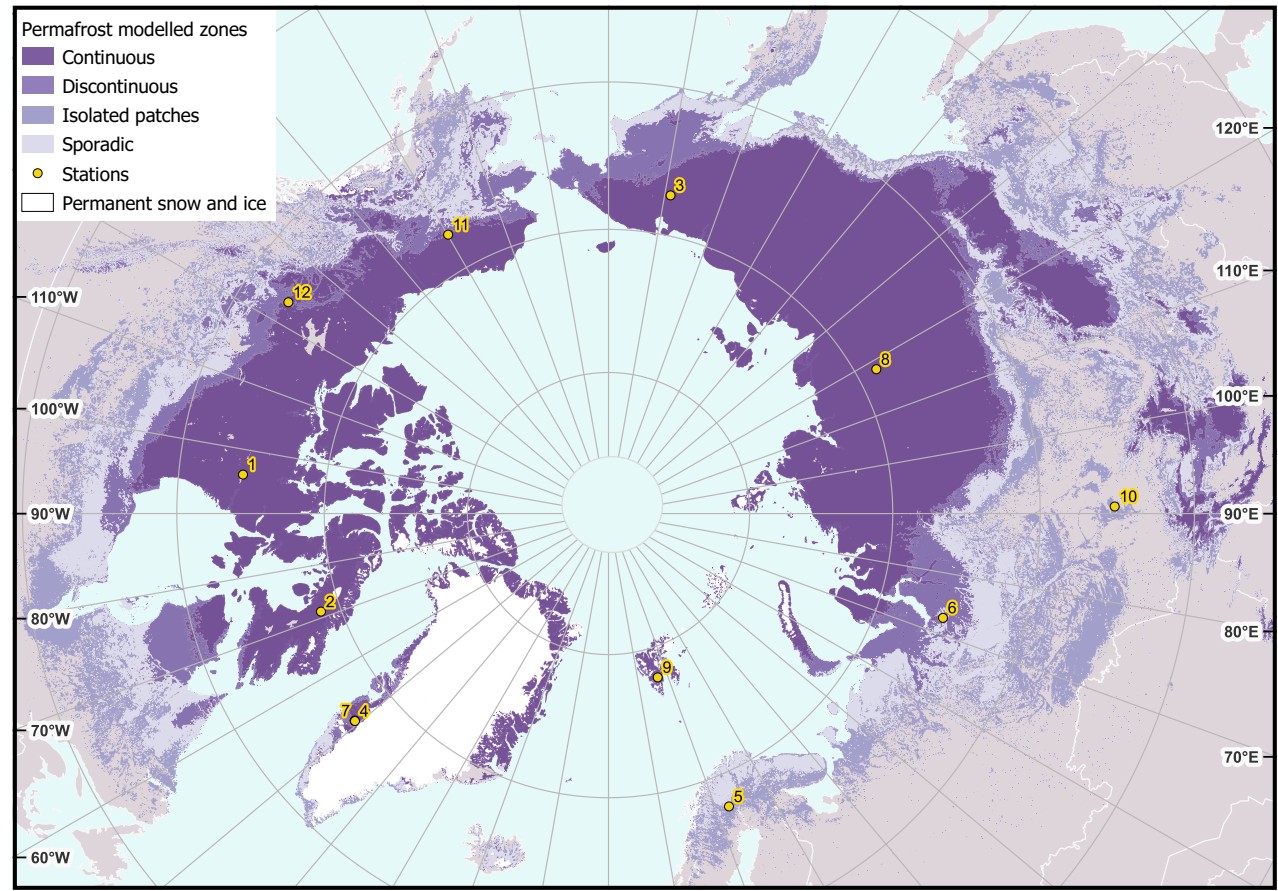

**Figure 1.** Selected EUSTACE stations with their ID (Table 1). Permafrost zonation map adapted from Obu et al. (2018).



**Table 1.** Selected EUSTACE meteorological stations with WMO code, geolocation, elevation and available period.

| Station ID | Station code | Station name | Latitude | Longitude | Elevation | Start year | End year |
|---|---|---|---|---|---|---|---|
| 1 | CA002300500 | BAKER LAKE A | 64.30 | -96.08 | 19 | 1946 | 2015 |
| 2 | CA002401030 | DEWAR LAKES | 68.65 | -71.17 | 527 | 1958 | 2015 |
| 3 | TX_SOUID148829 | ILIRNEJ | 67.25 | 168.97 | 352 | 1944 | 2013 |
| 4 | TX_SOUID147048 | KANGERLUSSUAQ | 67.02 | -50.70 | 50 | 1975 | 2015 |
| 5 | TX_SOUID137416 | LAINIO | 67.76 | 22.35 | 315 | 1965 | 2015 |
| 6 | TX_SOUID148484 | NADYM | 65.47 | 72.67 | 14 | 1959 | 2013 |
| 7 | GLW00016504 | SONDRESTROM | 67.02 | -50.80 | 50 | 1947 | 2015 |
| 8 | TX_SOUID148639 | SUHANA | 68.62 | 118.33 | 78 | 1938 | 2013 |
| 9 | TX_SOUID111376 | SVEAGRUVA | 77.88 | 16.72 | 9 | 1978 | 2015 |
| 10 | TX_SOUID150449 | SVETLOLOBOVO | 55.10 | 90.80 | 326 | 1958 | 2013 |
| 11 | USC00509869 | WISEMAN | 67.42 | -150.11 | 349 | 1918 | 2015 |
| 12 | CA002204000 | WRIGLEY A | 63.22 | -123.43 | 150 | 1943 | 2013 |

## 2.4 Auxiliary data

The generation of LST data requires auxiliary datasets:

- Skin temperature (Tskin) and Total Column Water Vapor (TCWV) from the MERRA-2 reanalysis dataset (M2T1NXSLV, variables are labelled TS and TQV). The data come at hourly temporal resolution with a spatial resolution of 0.5° x 0.625°. Nearest neighbour resampling was performed to match the AVHRR spatial resolution and scanline time, i.e. as in the work of Ma et al. (2020).

- ESA CCI Land Cover 1992-2015 and Copernicus ICDR Land Cover 2016-2020 datasets are used. Both datasets are consistent with each other. Their spatial resolution was decreased to match AVHRR GAC spatial resolution. The class labels follow the Land Cover Classification System (LCCS) developed by the United Nations (UN) Food and Agriculture Organization (FAO) (Copernicus Climate Change Service, Climate Data Store, (2019)).

- Atmospheric profiles from the Clear-Sky Database developed at LSA-SAF (Ermida and Trigo, 2022) are used for the RT modelling (RTM). This database contains atmospheric profiles such as temperature, specific humidity and ozone on 137 model levels (full vertical resolution), sampled from ERA5 for the 2009-2019 period. The sampling technique follows the method from Chevallier et al. (2000). Surface variables like T2M, surface pressure, Tskin and emissivity are obtained from the combination of ERA5 and satellite data. Column variables, such as TCWV and total cloud cover (TCC) are also present in the database. The atmospheric profiles are classified on TCWV varying from 0 to 60 mm and TS ranging from 190 to 340 K.



   – In situ LST measurements from the Surface Radiation Budget (SURFRAD) network (https://gml.noaa.gov/dv/data/index.php?parameter_name=Surface%2BRadiation) and Karlsruhe Institute of Technology (KIT) network (https://www.imk-asf.kit.edu/english/skl_surfacetemperature.php) (Göttsche et al., 2016; Martin et al., 2019) are used for validation. Table 2 lists the stations. The KIT stations, which are part of LSA SAF's validation effort and supported by EUMETSAT, are located in different climate zones (Göttsche et al., 2016).

   – The Copernicus digital elevation model (DEM) GLO-90 upscaled to 0.05° spatial resolution is used (https://doi.org/10.5270/ESA-c5d3d65) for the RT modelling. This dataset represents the surface of the Earth and is based on radar satellite data obtained from the TanDem-X Mission.

**Table 2.** Description of the stations used for LST validation. Station name and ID, the network the station belongs to, latitude, longitude, elevation and the dominant land cover type are listed.

| Station name (ID) | Network | Latitude | Longitude | Elevation [m] | LCCS |
|---|---|---|---|---|---|
| Bondville, Illinois (BND) | SURFRAD | 40.0519 | -88.3731 | 230 | Cropland |
| Desert Rock, Nevada (DRA) | SURFRAD | 36.6237 | -116.0195 | 1007 | Open Shrubland |
| Fort Peck, Montana (FPK) | SURFRAD | 48.3078 | -105.1017 | 634 | Grassland |
| Goodwin Creek, Mississippi (GCM) | SURFRAD | 34.2547 | -89.8729 | 98 | Wooded Grassland |
| Penn. State Univ., Pennsylvania (PSU) | SURFRAD | 40.7201 | -77.9309 | 376 | Deciduous Broadleaf Forest |
| Sioux Falls, South Dakota (SFA) | SURFRAD | 43.73403 | -96.62328 | 1689 | Cropland |
| ARM Southern Great Plains, Oklahoma (SGP) | SURFRAD | 36.60406 | -97.48525 | 314 | Cropland |
| Table Mountain, Boulder, Colorado (TBL) | SURFRAD | 40.1250 | -105.2368 | 1689 | Cropland |
| Lake Constance, Germany (BOD) | KIT | 47.58 | 9.57 | 396 | Water |
| Evora, Portugal (EVO) | KIT | 38.54 | -8.003 | 300 | Mosaic Tree and Shrubs |

## 3 Methods

LST can be retrieved from thermal infrared data with the well-established split window (SW) method (Ma et al., 2020; Yang et al., 2020; Reiners et al., 2021). Since 1983 (Price, 1984; Prata, 1994) different algorithms have been developed to obtain LST



as a function of the satellite-recorded brightness temperature (BT). The split-window approach takes advantage of the different water vapour absorption characteristics of two adjacent channels (Lieberherr et al., 2017; Ma et al., 2020). LST is affected by many factors, which requires additional terms to model the effects of land cover type, viewing angle and topography (Trigo et al., 2009).

## 3.1 Generalised Split Window algorithm

For this study the Generalized Split Window (GSW) algorithm, developed by Wan and Dozier (1996) (Eq. 1) and used within the framework of LSA-SAF (Trigo et al., 2008b; Ermida and Trigo, 2022) is selected. The GSW performs well on a global scale and has the highest relative accuracy among a selection of SW algorithms investigated in the work of Yang et al. (2020). The GSW depends on channel-effective emissivity and sensor-specific coefficients that must be determined for the expected atmospheric and surface conditions.

$$T_s = (A_1 + A_2\frac{1-\epsilon}{\epsilon} + A_3\frac{\Delta\epsilon}{\epsilon^2})(T_{11} + T_{12}) + (B_1 + B_2\frac{1-\epsilon}{\epsilon} + B_3\frac{\Delta\epsilon}{\epsilon^2})(T_{11} - T_{12}) + C \qquad (1)$$

where $T_{11}$ and $T_{12}$ denote BT of channels centred at approximately 11 and 12 µm, $\epsilon = (\epsilon_{11} + \epsilon_{12})/2$, $\Delta\epsilon = (\epsilon_{11} - \epsilon_{12})$, $A_i$, $B_i$ and $C$ are split window coefficients (SWC).

The split-window coefficients in Eq.1 are obtained by applying multi-linear regression on a set of simulated BTs against a calibration database. Simulated BTs are obtained by performing RTM with version 13 of the Radiative Transfer for TOVS (RTTOV) developed at NWC SAF (Saunders et al., 2018). The python wrapper was used in this study (Hocking et al., 2021). Atmospheric profiles, surface and column variables from the clear-sky profile database developed by Ermida and Trigo (2022) complemented with elevation information and satellite viewing angles, are ingested by RTTOV. The clear-sky profile database contains 97 files, each file containing approximately 1000 atmospheric profiles corresponding to a different class of TCWV and Tskin. In total, the database contains 82'793 profiles, and each profile possesses six different TS values and 25 pairs of emissivity at 11 µm and 12 µm. More details can be found in Ermida and Trigo (2022). Convolving the TOA radiances produced by RTTOV with the specific AVHRR response functions yields BTs as seen by the different sensors for different atmospheric and viewing conditions. RT modelling was performed on the calibration dataset for each satellite and 15 different satellite view zenith angles (VZA) ranging from 0 to 70°. Table 3 summarises the construction of the simulation dataset. Finally, for each class, the sample was split into a training (70%) and test (30%) set, and multilinear regression was performed on the resulting BTs. Based on the test sets, look-up tables (LUT) with coefficients are created for each satellite. Mean absolute error (MAE), the coefficient of determination ($R^2$) and root mean square error (RMSE) are computed for all coefficients to keep track of the general performance of the RTM.

**Table 3.** Summary of the simulation dataset: the number of profiles used and the number of instances of view zenith angles (VZA), TS and LSE are shown.

| Source | Number of profiles | # VZA | # TS | # LSE | Sample size |
| --- | --- | --- | --- | --- | --- |
| LSA SAF (Ermida and Trigo, 2022) | 82,793 | 15 | 6 | 25 | 186,284,250 |



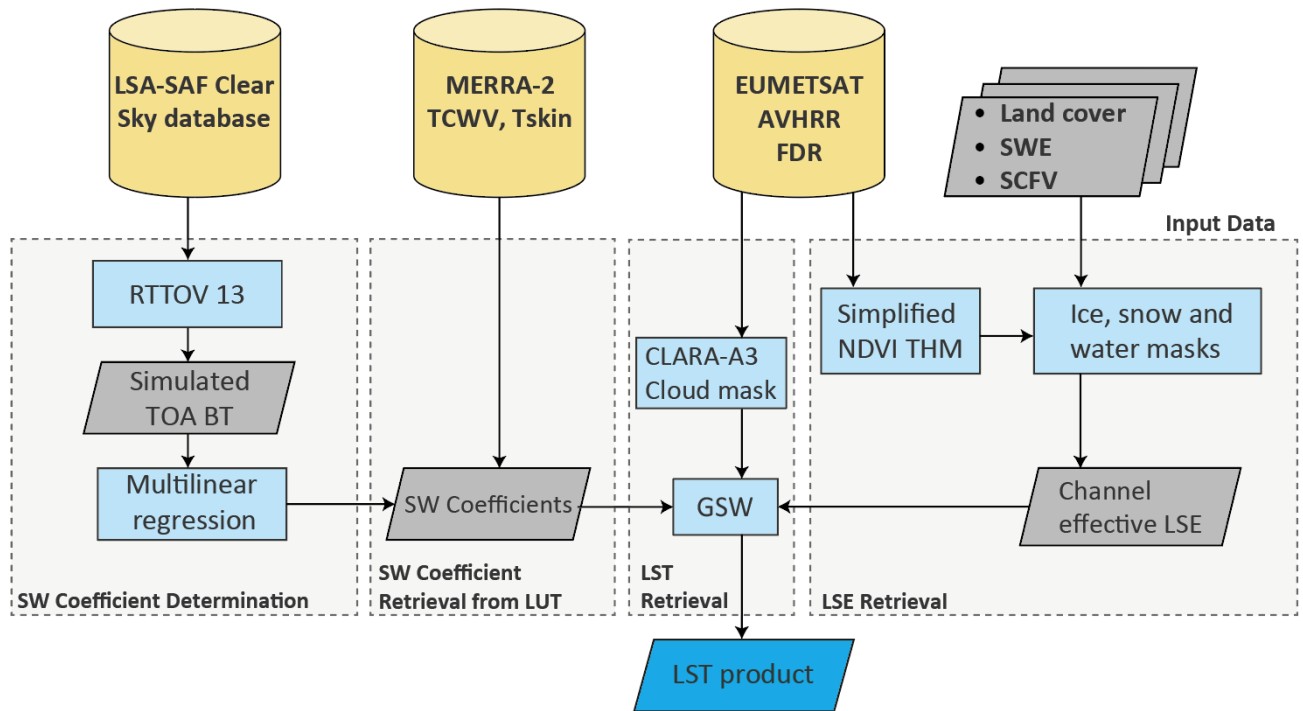

**Figure 2.** LST retrieval from AVHRR data. Input data are shown at the top. The leftmost box refers to the SW coefficient determination; the next box on the right displays the look-up-table storing coefficients for different atmospheric and surface conditions. The second box from the right presents the cloud mask application to the thermal channels and the rightmost panel the process to generate Land Surface Emissivity (LSE) based on land cover and NDVI information.

### 3.1.1 Land surface emissivity retrieval

Land surface emissivity (LSE) is retrieved by combining the simplified Normalized Difference Vegetation Index (NDVI) Threshold Method ($SNDVI_{THM}$) (Sobrino et al., 2008) based on (Sobrino and Raissouni, 2000) with channel emissivity data from spectral libraries and static land cover classifications.

First, 10-day NDVI maximum value composites (MVC) are generated from the AVHRR channels 1 and 2. NDVI thresholds that determine if a pixel is considered fully vegetated or entirely bare soil are set. In the present case, $NDVI_{soil}$ is set to 0.2, and $NDVI_{veg}$ is set to 0.5 (Sobrino et al., 2001). All pixels that have $0.2 <$ NDVI $< 0.5$ are considered mixed pixels and the corresponding emissivity is obtained by using the proportion of vegetation ($P_v$) method (Sobrino et al., 2008) (Eq. 2) that weighs the emissivity of bare soil ($\epsilon_{si}$) and vegetation ($\epsilon_{vi}$) for AVHRR channel $i$ ($i$=4 or 5).

$\epsilon_i = \epsilon_{vi} P_v + \epsilon_{si}(1 - P_v)$ \hfill (2)





$\epsilon_{si}$ and $\epsilon_{vi}$ are taken from a LUT based on information from spectral libraries (Trigo et al., 2008a; Peres and DaCamara, 2005). The emissivity of pixels with NDVI < 0.2 or NDVI > 0.5 is set to $\epsilon_{si}$ and $\epsilon_{vi}$, respectively. Here, the channel emissivities from Trigo (2008b, Table I) are used. The IGBP classes (Sulla-Menashe and Friedl, 2018) are converted to land cover classes of the ESA CCI project with plant functional types look-up tables (Wang et al. 2023). From 1992 to 2020, the ESA CCI land

cover at the 12 selected stations changed very little. Therefore, to reduce emissivity uncertainties due to unknown land cover information before 1992, a static land cover from 2000 is used throughout the project (Freitas et al., 2010). The land cover was previously upscaled to the resolution of the AVHRR dataset. $P_v$ is obtained from NDVI with Eq. 3 (Carlson and Ripley, 1997).

$$P_v = \left( \frac{NDVI - NDVI_{soil}}{NDVI_{veg} - NDVI_{soil}} \right)^2 \tag{3}$$

Pixels with low NDVI values (NDVI < 0.2) are defined as bare soil. Such NDVI values can also indicate the presence of

snow or cloud cover. Snow cover extent information is retrieved from two data products from the ESA CCI+ Snow project and cloudy pixels are masked out in the final LST product. A threshold of 70 % of SCFV or 4 mm for SWE is used to categorise the pixel as fully snow-covered. Snow covered pixels are assigned to laboratory emissivity spectra values of medium snow (Fig. 5 of Hulley et al. (2014)). Water pixels and permanent snow and ice areas are retrieved based on land cover information from the ESA Landcover CCI for the year 2000 and are assigned to channel effective emissivity values from Hulley et al. (2014).

**3.2 LST Retrieval**

LST is retrieved as follows: (i) all necessary data (BTs, cloud mask, emissivity and atmospheric data) are read, (ii) atmospheric data from MERRA-2 (TS and TQV) are downscaled to GAC spatial resolution, the corresponding timestamp is matched with the scanline time for each pixel, (iii) based on satellite identification number, satellite viewing angle, total column water vapour (TQV) and skin temperature (TS) and the corresponding SWC from the LUT are assigned to each pixel and (iv) LST is

computed from channel BTs, emissivities and the assigned SWC (Eq. 1). Pixels with a satellite view zenith angle greater than 40° and the MAE of the test set in the RTM simulations greater than 0.5 K are masked out.

**3.3 Validation Procedure**

The AVHRR LST dataset is validated against in situ data from different sites (Table 2). In situ LST and AVHRR LST datasets are joint based on acquisition time and geolocation. The closest pixel to the station is taken, and a time difference of up to 5

minutes between the satellite overpass and the in situ measurement is considered. For Lake Constance, a time difference of up to 30 minutes is allowed. Similarly, as in Ma et al. (2020), three-sigma filtering was performed to remove outliers (Pearson, 2002). The most accurate surface temperatures can be obtained over large water bodies, such as lakes and reservoirs. Densely vegetated surfaces are also particularly suitable for LST validation (Coll et al., 2009).



### 3.4 LST AVHRR time series generation

Depending on the heterogeneity of the land cover, between four and nine AVHRR LST GAC pixels are extracted around each station. Pixels that have a cloud fraction higher than 0.1 are removed, and the average of the remaining pixels is computed. Data from NOAA-7, 9, 11, 14, 16, 18 and 19, as well as the entire MetOp series, are considered for constructing the time series. The considered period for each satellite is chosen to minimise orbital drift and avoid the outage periods (EUMETSAT, 2023d). The retained periods are listed in Table 4.

**Table 4.** Considered time period for each satellite and sensor that it carries.

| Satellite | Platform | Valid Period |
| --- | --- | --- |
| NOAA-07 | AVHRR-2 | 24 November 1981 — 01 February 1985 |
| NOAA-09 | AVHRR-2 | 25 February 1985 — 07 November 1988 |
| NOAA-11 | AVHRR-2 | 08 November 1988 — 16 October 1994 |
| NOAA-14 | AVHRR-2 | 20 January 1995 – 31 December 2000 |
| NOAA-16 | AVHRR-3 | 01 January 2001 – 30 June 2005 |
| NOAA-18 | AVHRR-3 | 01 July 2005 – 28 February 2009 |
| NOAA-19 | AVHRR-3 | 01 March 2009 – 31 December 2015 |
| METOP-A | AVHRR-3 | 01 January 2016 – 31 December 2018 |
| METOP-B | AVHRR-3 | 01 January 2016 – 31 December 2020 |
| METOP-C | AVHRR-3 | 03 July 2019 — 31 December 2020 |

Once the relevant periods are extracted, outlier detection is performed based on a 10-day rolling window analysis and detected outliers are removed. Daily temperature variability is very high (Mildrexler et al., 2011), and AVHRR-derived LST time series are subject to noise, therefore, monthly means are computed from the merged time series for further analysis.

### 3.5 Time series analysis

The AVHRR LST monthly means time series is compared with two independent data sets: ERA5-Land and EUSTACE Tair.
ERA5-Land pixels collocated to each weather station are selected, and values of T2M for the same position are extracted. The EUSTACE Tair data are filtered to keep only data that passed all quality assurance checks (Menne et al., 2012), and monthly means are computed. This yields time series of LST, ERA5-Land T2M (hereafter referred to as 'T2M') and EUSTACE Tair (hereafter referred to as 'Tair') for each station. Before performing the actual stability analysis and starting with trend analysis, the overall relationship between the datasets is assessed by computing statistical parameters such as the Pearson correlation
coefficient (r), the MAE and the RMSE. Previous studies (Mildrexler et al., 2011; Hachem et al., 2012; Westermann et al., 2012; Urban et al., 2013) found variability in the LST-Tair correlation depending on the season and the land cover.

Then, anomalies of the monthly mean time series are computed for the LST, T2M and Tair time series at all stations to remove the strong seasonal cycle inherent to temperature data (Good et al., 2022). A temperature anomaly describes the difference from





a baseline climatology. In the present study, the anomalies are computed by subtracting the mean temperature for the entire

time series from the monthly values. The LST, T2M and Tair anomalies are compared by computing the Pearson correlation

coefficient (r). Three different periods are considered for the anomaly analysis: a) the entire year, b) polar winter (December,

January) and c) polar summer (June and July). June and July are chosen as summer months to respect the symmetry of the

winter period. First, the relationship between the anomalies of the different datasets is evaluated with the Pearson correlation

coefficient (r) and the general stability of the LST dataset is assessed by computing the trends of the anomaly differences.

Three sets of differences are computed as:

$$\Delta Tanom_{LST-T2M} = LST_{anom} - T2M_{anom} \tag{4}$$

$$\Delta Tanom_{LST-Tair} = LST_{anom} - Tair_{anom} \tag{5}$$

$$\Delta Tanom_{Tair-T2M} = Tair_{anom} - T2M_{anom} \tag{6}$$

The stability analysis is performed separately on the summer and winter periods. Non-parametric trend analysis is per-

formed on the anomaly differences using the Thiel-Sen slope to quantify the trend and the Mann-Kendall test to determine

its significance. The Python implementation of the Mann-Kendall Trend Test (Hussain and Mahmud, 2019) was used in this

work. Finally, trends are computed at each station for the entire year for all three datasets, and summer and winter trends are

computed for the whole Pan-Arctic region.

## 4  Results

In the first parts of the Results section (Sect. 4.1.1 and 4.1.2) the performances of the GSW algorithm and the validation results

are presented. The remaining sub-sections present the relationships of the different datasets as well as the stability analysis and

trend computation.

### 4.1  LST validation results

#### 4.1.1  Performance of the split window algorithm

Multi-linear regression was used to fit Eq. 1 to the RTM results for each class of TCWV and Tskin in a training set that

consisted of 70 % of the samples, thereby retrieving the corresponding SWCs. The performance of the regression model was

evaluated with the remaining independent samples (test set). The performance of the model is assessed by using the MAE

and the coefficient of determination ($R^2$). The values shown in Fig. 3 are mean values across all satellites: the MAE of the



predictions are below 0.5 kelvin for dry conditions (TCWV< 30) and low satellite viewing angle (VZA < 40). In cases with
very moist atmospheres and high VZA (VZA> 50), the MAE increases substantially, and $R^2$ is considerably lower ( $R^2$< 0.92).
Higher temperatures and higher VZA also lead to an increased error. The overall MAE is always below two K for water vapour
below 50 mm TCWV. MAE values above 2 K are reached for high TCWV (TCWV > 50 mm) and high VZA values (VZA >
50 °). Overall, the amount of water vapour in the Arctic atmosphere is low, which indicates that the model is well suited for the
present use case.

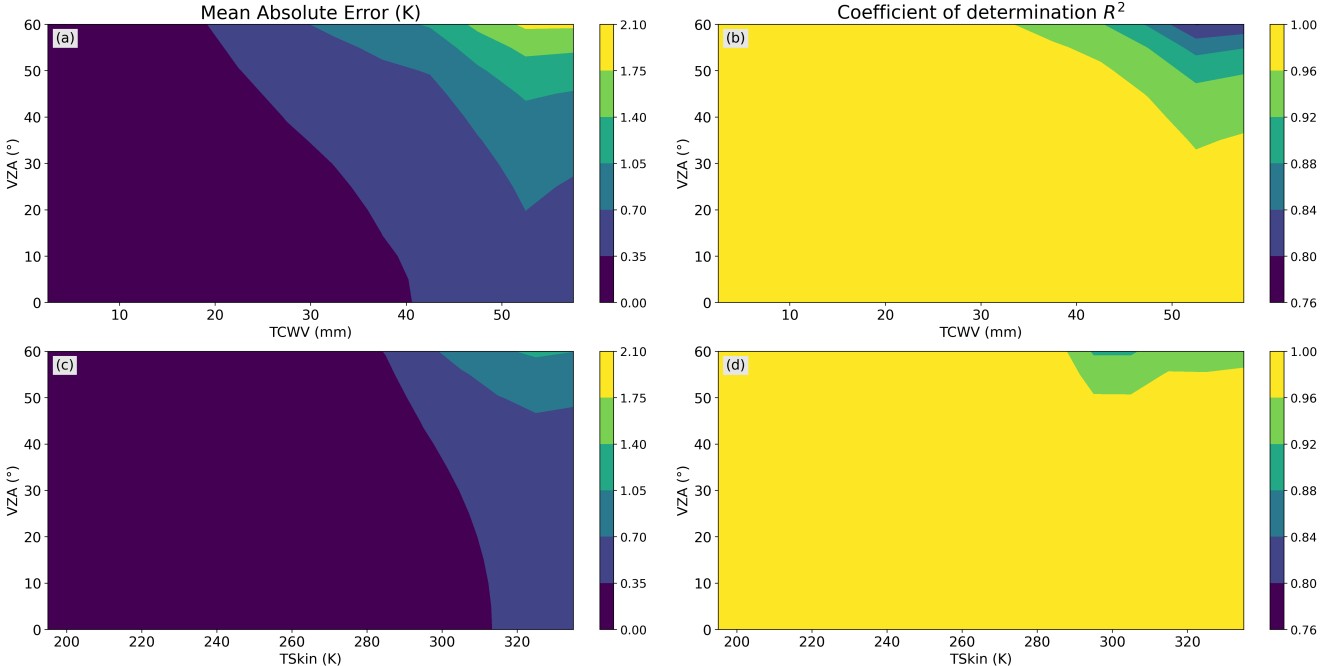

**Figure 3.** SWA performances: (a) distribution of the mean absolute error (MAE) and (b) coefficient of determination as a function of the
view zenith angle (VZA) and total column water vapour (TCWV). (c) and (d) show the corresponding distributions in dependence of VZA
and Tskin.

### 4.1.2 Validation with in situ LST

Fig. 4 shows the validation results for NOAA-14, 16, 17, 18 and 19, and MetOP-A, B and C against in situ LST from the
validation sites in Table 2. The match-up with the SURFRARD stations covers the period from 1985 to 2020, the match-
up period for the KIT station in EVORA (EVO) starts in 2009 and ends in 2020, and finally, the match-up period for Lake
Constance starts in 2016 and ends in 2020. EVO is located in an evergreen oak woodland with approximately 33% of tree
295 crown cover, which can affect the satellite-retrieved LST due to directional effects (Guillevic et al., 2013; Ermida et al., 2014).
Therefore, in Evora, only nighttime data are considered for the validation. The in situ data for Lake Constance (BOD) are
collected during the operating hours of the ferry and are thus only available during daytime. Both daytime and nighttime data
are considered at the SURFRAD sites. BOD has the fewest available points due to the shorter match-up period. The overall



RMSE range for the ten stations is 0.80-3.43 K. The highest performance is reached at BOD. LST are more stable over water

300   bodies, and water emissivity is less prone to induce significant uncertainties (Masuda et al., 1988; Niclòs et al., 2005). On land, the highest agreement was obtained for the Sioux Falls (SFA) and Goodwin Creek (GCM) sites. The lowest agreement was found at Desert Rock (DRA). This poor performance may result from the desert land cover at that station and the chosen emissivity retrieval method that performs better on vegetated surfaces (Trigo et al., 2008a). Compared to previous studies on AVHRR LST (Ma et al., 2020; Reiners et al., 2021; Li et al., 2023a), the present dataset shows a similar accuracy and precision.







**Figure 4.** AVHRR LST versus in situ LST at (a) Bondville (BND), (b) Desert Rock (DRA), (c) Fort Peck (FPK), (d) Goodwin Creek (GCM), (e) Penn. State Univ (PSU), (f) Sioux Falls (SFA), (g) Southern Great Plains (SGP), (h) Table Mountain (TBL), (i) Evora (EVO), (j) Lake Constance (BOD). Match-up periods are provided in the text.



## 4.2 Relationships between the temperature datasets

The relationship between AVHRR LST and station EUSTACE Tair, and AVHRR LST and ERA5-Land T2M at each EUSTACE station is first assessed by computing the Pearson correlation coefficient (r), the MAE and RMSE of the monthly mean temperature data (see Sect. 3.5). Comparisons against $Tair_{min}$ show, in general, higher RMSE and MAE values than comparisons against $Tair_{max}$ and T2M (Table 5). For the stations in Table 5, the mean MAE values are 8.47, 5.56 and 5.92 K for LST versus -$Tair_{min}$, -$Tair_{max}$ and -T2M respectively. Stations WRIGLEY A, WISEMAN and SVEGRUVA exhibit lower MAE and RMSE values than the other stations. WISEMAN and SVEGRUVA have slightly fewer comparison samples (N<230). SVEAGRUVA, located in Svalbard, is the northernmost station and experiences persistent cloud cover, leading to fewer usable satellite observations. The EUSTACE time series at WISEMAN is shorter due to missing data. WRIGLEY A exhibits low MAE and RMSE values when compared to $Tair_{max}$ but presents similar values to other stations when compared to $Tair_{min}$.

As to be expected, the analysis reveals a high degree of correlation between monthly LST, Tair and T2M with correlation coefficients all above 0.9 (r) (Table 5). In line with higher RMSE and MAE, $Tair_{min}$ has slightly lower (r) values when compared to LST. The better performance of $Tair_{max}$ can be attributed to the closer daytime overpass of the NOAA/METOP satellites (Good et al., 2022). In opposition, $Tair_{min}$ is generally recorded during night.

**Table 5.** MAE, RMSE, and Pearson Coefficient (r) at the selected EUSTACE stations for monthly mean comparisons. N is the number of samples.

| | **Relationship (MAE, RMSE, and (r)) of monthly means** | | | | | | | | | |
| Station name | **LST versus Tair min** | | | **LST versus Tair max** | | | **LST versus T2M** | | | N |
| | **RMSE** | **MAE** | **r** | **RMSE** | **MAE** | **r** | **RMSE** | **MAE** | **r** | |
| BAKER LAKE A | 9.56 | 8.07 | 0.97 | 5.27 | 4.26 | 0.98 | 6.40 | 4.89 | 0.98 | 298 |
| DEWAR LAKES | 10.62 | 8.37 | 0.95 | 9.92 | 8.46 | 0.96 | 8.17 | 6.18 | 0.97 | 330 |
| ILIRNEJ | 10.06 | 8.58 | 0.97 | 5.92 | 4.98 | 0.98 | 7.65 | 6.2 | 0.98 | 316 |
| KANGERLUSSUAQ | 10.80 | 8.65 | 0.95 | 8.69 | 7.28 | 0.96 | 8.38 | 6.91 | 0.97 | 357 |
| LAINIO | 7.78 | 6.61 | 0.96 | 5.34 | 4.18 | 0.97 | 5.63 | 4.59 | 0.97 | 290 |
| NADYM | 9.52 | 8.05 | 0.97 | 6.71 | 5.33 | 0.98 | 7.41 | 6.06 | 0.98 | 282 |
| SONDRESTROM | 12.15 | 9.91 | 0.95 | 8.11 | 6.93 | 0.96 | 9.39 | 7.76 | 0.97 | 329 |
| SUHANA | 9.82 | 8.70 | 0.98 | 5.76 | 4.62 | 0.98 | 6.37 | 5.20 | 0.98 | 339 |
| SVEAGRUVA | 5.03 | 3.66 | 0.93 | 7.17 | 5.82 | 0.94 | 5.65 | 4.22 | 0.94 | 228 |
| SVETLOLOBOVO | 15.30 | 13.08 | 0.96 | 7.31 | 6.06 | 0.97 | 10.72 | 8.94 | 0.97 | 343 |
| WISEMAN | 9.42 | 7.98 | 0.96 | 6.79 | 5.54 | 0.98 | 6.43 | 5.20 | 0.98 | 176 |
| WRIGLEY A | 10.79 | 9.97 | 0.97 | 4.39 | 3.31 | 0.98 | 5.75 | 4.91 | 0.98 | 279 |




The relationship between the monthly AVHRR LST anomalies versus EUSTACE Tair anomalies and ERA5-Land T2M anomalies is assessed with the Pearson correlation coefficient (r) (Table 6). All stations except SUHANA (Siberia) and SVEA-GRUVA (Svalbard) display strong positive correlations (r > 0.5) between LST and both air temperature datasets (Tair and T2M). Correlations are consistent for both air temperature datasets; only minor differences are visible. The (r) values for comparing Tair with T2M are higher than those in the corresponding comparison with LST. Correlations of LST versus T2M anomalies vary between 0.46 (r) and 0.71 (r). For $Tair_{max}$ versus LST, the (r) values are between 0.40 and 0.71, whereas for $Tair_{min}$ (r) has a range of 0.35-0.70. In general, slightly higher (r) values are obtained for the comparison against T2M than Tair. Comparison values of Tair versus T2M have values (r) between 0.69 and 0.97. Similarly than in Table 5, lower (r) values are found for the comparison with $Tair_{min}$ than for $Tair_{max}$. The lowest correlation value between T2M and Tair is obtained at WISEMAN (Alaska). NADYM (Russia) shows consistently high correlations (r ≈ 0.7) across all comparisons and has the highest correlation value for the Tair versus T2M evaluation. Previous studies have also found high correlations between station Tair and LST data for other LST datasets (Urban et al., 2013; Good et al., 2022), e.g. from the ESA CCI project (Ghent et al., 2023). The differences of (r) between Table 5 and Table 6 can be attributed to a phase shift between the anomalies and to a strong seasonal signal present in the time series on monthly means.

**Table 6.** Correlation Coefficient (r) from all stations for the comparison between the LST monthly mean anomalies versus the T2M and Tair monthly mean anomalies.

| Station name | Pearson correlation coefficient (r) of the monthly mean anomalies | | | | |
| --- | --- | --- | --- | --- | --- |
| | LST vs Tair min | LST vs Tair max | LST vs T2M | T2M vs Tair min | T2M vs Tair max |
| BAKER LAKE A | 0.53 | 0.56 | 0.57 | 0.91 | 0.92 |
| DEWAR LAKES | 0.59 | 0.58 | 0.63 | 0.88 | 0.89 |
| ILIRNEJ | 0.56 | 0.61 | 0.61 | 0.86 | 0.91 |
| KANGERLUSSUAQ | 0.66 | 0.68 | 0.70 | 0.93 | 0.96 |
| LAINIO | 0.61 | 0.65 | 0.64 | 0.93 | 0.94 |
| NADYM | 0.70 | 0.71 | 0.71 | 0.97 | 0.97 |
| SONDRESTROM | 0.67 | 0.70 | 0.69 | 0.93 | 0.96 |
| SUHANA | 0.43 | 0.48 | 0.51 | 0.88 | 0.94 |
| SVEAGRUVA | 0.35 | 0.40 | 0.46 | 0.92 | 0.95 |
| SVETLOLOBOVO | 0.59 | 0.70 | 0.68 | 0.91 | 0.95 |
| WISEMAN | 0.47 | 0.61 | 0.65 | 0.69 | 0.75 |
| WRIGLEY A | 0.57 | 0.57 | 0.62 | 0.85 | 0.87 |

To assess the general stability of the LST dataset, the differences between the monthly anomalies of the datasets (Eq. 4, Eq. 5 and Eq. 6) and the trends of these differences are calculated. Since high correlation values were obtained for T2M and $Tair_{max}$, these two datasets are considered for the LST stability analysis. The confidence interval is set to 95%, meaning that trends with





p-values below 0.05 present a significant trend. Ideally, the trend of the difference should be zero or very close to zero. The results are shown in Table 7: ten of the 24 trends of the anomaly differences involving LST present statistically significant trends. This indicates that for these datasets, the difference between the anomalies increases or decreases over time and, thus, is not stable. Four stations (DEWAR LAKES, SVEAGRUVA, SVETLOLOBOVO and WISEMAN) show statistically significant

trends when comparing T2M and $Tair_{max}$ anomalies, e.g. see SVEAGRUVA in Fig. 5. However, these stations present stable trends for the comparison with LST anomalies. For example, the comparison of LST versus T2M at SEVAGRUVA is very stable (0.10 K/decade) (see Fig. 5). The same observation can be made at DEWAR LAKES. SVEAGRUVA is a special case as the data are very sparse. Overall, the LST - $Tair_{max}$ trend and LST - T2M trends shown in Table 7 are not consitent accross datasets. For example, Fig. 6 shows the trends of the anomalies at KANGERLUSSUAQ. All trends are stable (statistically non-

significant), but differences are visible in the trend values for both LST comparisons. Plots of the trends of the other stations are shown in Appendix A1. Several reasons, such as orbit drift, missing data or corrupt station data, could explain the observed discrepancies. Instabilities are also visible in the Tair versus T2M comparisons. The significant trend for the comparison of Tair versus T2M can be explained by missing station data (appendix A1). Six of the 12 LST versus T2M experiments are statistically non-significant, suggesting no detectable trends for the anomaly differences at these stations. Regarding the

$Tair_{max}$ anomalies minus LST anomalies trends, four out of 12 stations show a significant trend. The stations located at DEWAR LAKES, KANGERLUSSUAQ, LAINIO, SUHANA, SVEAGRUVA and WRIGLEY A do not show a significant trend in the comparisons to LST (see Table 7).

**Table 7.** Trend of the anomaly differences for the three pairs of differences. Trends in italic are significant (p-value < 0.05) and associated p-values are marked in bold.

| Station name | LST-T2M | | LST-Tair max | | Tair max-T2M | |
|---|---|---|---|---|---|---|
| | **Trend [K/dec.]** | **P-value** | **Trend [K/dec.]** | **P-value** | **Trend [K/dec.]** | **P-value** |
| BAKER LAKE A | *0.52* | **0.0005** | *0.63* | **0.0003** | -0.07 | 0.13 |
| DEWAR LAKES | 0.04 | 0.81 | 0.22 | 0.22 | *-0.11* | **0.03** |
| ILIRNEJ | *0.35* | **0.005** | 0.21 | 0.21 | 0.03 | 0.72 |
| KANGERLUSSUAQ | -0.23 | 0.13 | -0.10 | 0.62 | -0.04 | 0.42 |
| LAINIO | 0.25 | 0.054 | 0.10 | 0.50 | 0.06 | 0.08 |
| NADYM | *-0.57* | **0.003** | *-0.59* | **0.01** | -0.00 | 0.96 |
| SONDRESTROM | *0.37* | **0.04** | *0.55* | **0.008** | -0.05 | 0.35 |
| SUHANA | -0.16 | 0.27 | -0.33 | 0.06 | -0.01 | 0.80 |
| SVEAGRUVA | 0.10 | 0.63 | 0.35 | 0.10 | *-0.13* | **0.000** |
| SVETLOLOBOVO | *-0.52* | **0.002** | *-0.36* | **0.04** | *-0.17* | **0.005** |
| WISEMAN | *0.27* | **0.02** | 0.05 | 0.88 | *-0.56* | **0.006** |
| WRIGLEY A | 0.11 | 0.32 | 0.21 | 0.31 | -0.04 | 0.65 |



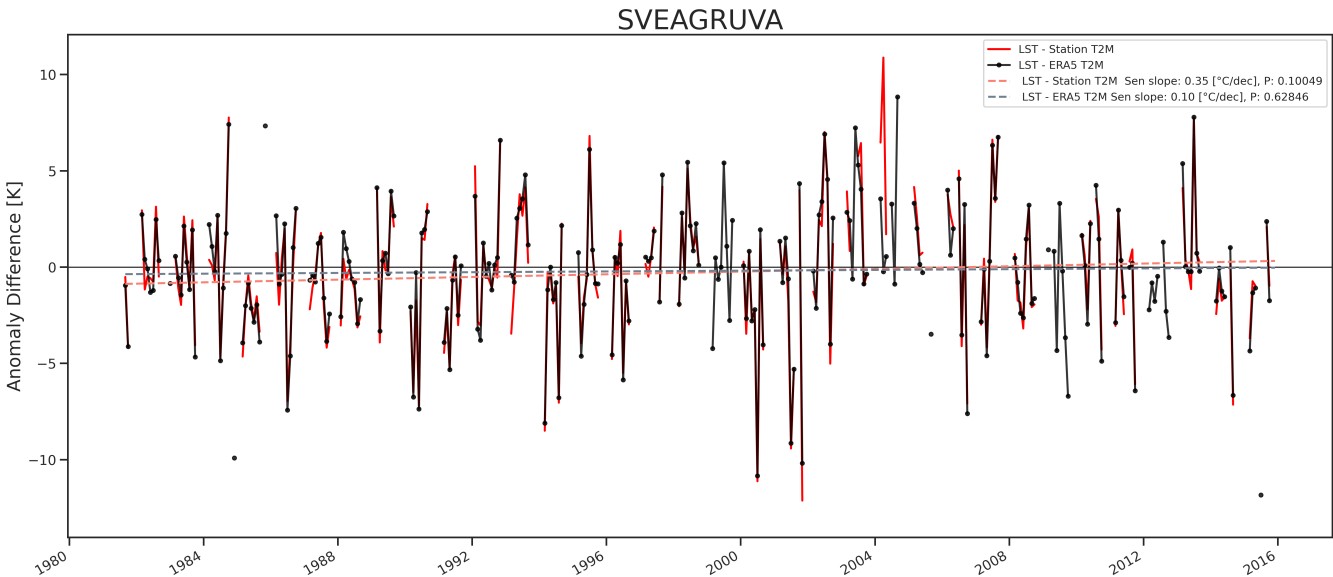

**Figure 5.** Monthly differences of the anomalies at SVEAGRUVA (Svalbard) between 1981 and 2015

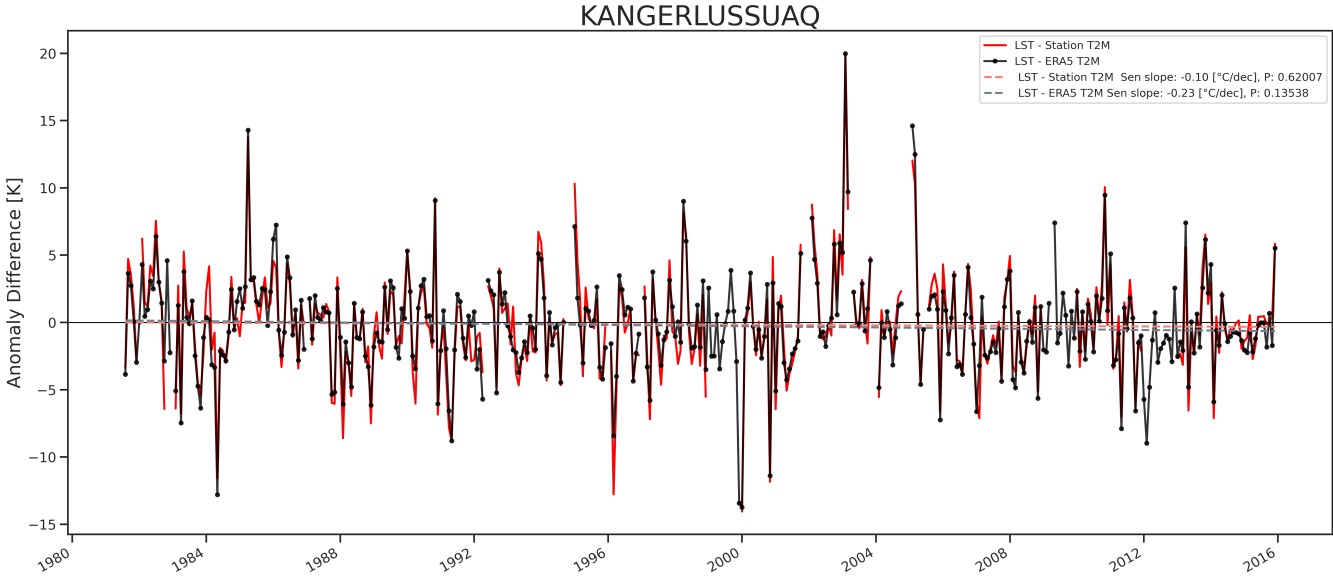

**Figure 6.** Monthly differences of the anomalies at KANGERKUSSUAQ (Greenland) between 1981 and 2015.



**Table 8.** Trends in monthly mean anomalies (K/decade) for selected stations. Values in italic are significant (p-value < 0.05) and associated p-values are marked in bold.

| Station name | LST | | EUSTACE Tair | | ERA5-Land T2M | |
|---|---|---|---|---|---|---|
| | Trend [K/dec.] | p-value | Trend [K/dec.] | p-value | Trend [K/dec.] | p-value |
| DEWAR LAKES | *0.61* | **0.005** | 0.21 | 0.11 | *0.39* | **0.002** |
| KANGERLUSSUAQ | *0.50* | **0.025** | *0.52* | **0.0008** | *0.55* | **0.0001** |
| LAINIO | *0.64* | **0.0004** | *0.64* | **0.0001** | *0.50* | **0.0001** |
| SUHANA | 0.26 | 0.18 | *0.47* | **0.007** | *0.54* | **0.0001** |
| WRIGLEY A | *0.47* | **0.009** | 0.34 | 0.1 | *0.32* | **0.005** |

LST and air temperature trends of stable stations (DEWAR LAKES, KANGERLUSSUAQ, LAINIO, SUHANA, SVEA-GRUVA and WRIGLEY A) have the same order of magnitude (Table 8) and all stations present a warming trend. DEWAR
LAKES, located in east of Nunavut (Canada), shows a more pronounced trend for LST than for the air temperature (Tair and T2M). Tair captured at the weather station does not present a significant trend. KANGERLUSSUAQ in Greenland shows similar trends in LST as in air temperature. LAINIO, located in a forested area in Northern Scandinavia, presents similar trends to LST and Tair, while the corresponding T2M trend is lower. The LST trend in SUHANA (Siberia) does not show a significant trend for LST, but both air temperature datasets show a consistent trend. WRIGLEY A, located in the Northwest Territories in
a forested area, shows a higher LST trend than T2M trend.

## 4.3 Analysis of Summer and Winter periods

To further investigate if the orbital drift of the NOAA satellites influences the stability analysis and determined trends, the previous analysis is now performed separately for polar winter and polar summer (see Sect. 3.5). Correlation coefficients (r), trends (slopes) and p-values of the differences between LST anomalies minus $Tair_{max}$ anomalies and LST anomalies minus
T2M anomalies are calculated.

Correlation coefficients (r) (Table 9) for summer and winter for the analysis against LST lie in the range 0.4-0.8. Correlation results are, on average, slightly higher for LST versus T2M than LST versus $Tair_{max}$, for winter and summer. SVEAGURVA had insufficient data points in winter, and the trends for summer were not significant. Therefore, this station was removed from the analysis. Winter shows only slightly higher correlation values than summer. At BAKER LAKE A, ILIRNEJ, SON-
DRESTROM and WISEMAN, (r) values in summer are higher than in winter. Correlation values are slightly higher for the separate seasons (Table 9) than for the general analysis (Table 6). Correlation results for the $Tair_{max}$ versus T2M experiment range between 0.57 and 0.96. The lowest correlation values are obtained at WISEMAN and WRIGLEY A. All of the stations except SVETLOLBOVO exhibit (r) values in a similar range in summer and winter. The air temperature correlation value at SVETLOLBOVO is considerably lower in summer than in winter. SVETLOLOBOVO is the station located the most in the
south at 55° latitude. At WRIGLEY A, (r) values are similar for all significant results.



The results in the previous section show that some instabilities could be detected from the trends of the anomaly differences. The same analysis is now performed separately for summer and winter (Table 10). Five out of 12 stations do not show any significant trend either in summer or winter. Except for SUHANA, the remaining stations only show a significant trend during summer. For example, KANGERLUSSUAQ (Table 10 and Fig. 7) presents a significant positive trend during summer but no

significant trend in winter. SUHANA experiences a significant positive trend in winter for the LST anomaly minus $Tair_{max}$ anomaly difference, but this trend is not observed in the comparison to T2M (Table 10 and Fig. 8). From Fig. 8, it can also be noticed that the differences between $Tair_{max}$ and T2M increase over time in winter and summer, which suggests discrepancies between air temperature measured at the station and from ERA5-Land. The magnitude of the LST anomalies is also generally higher than Tair and T2M magnitude (Fig. 7 and Fig. 8), which can be explained by the higher amplitude of the LST diurnal

cycle pattern compared to air temperature (Good, 2016; Sharifnezhadazizi et al., 2019). SVETLOLOBOVO and KANGER-LUSSUAQ present poor correlation results in summer (see Table 9) and also exhibit strong significant summer trends for the anomaly difference, contrary to winter trends that remain stable (Table 10).





**Table 9.** Pearson correlation coefficient (r) of the monthly anomalies for the summer and winter period.

| Station | | LST versus Tair max (r) | LST versus T2M (r) | Tair max versus T2M (r) |
|---|---|---|---|---|
| BAKER LAKE A | Summer | 0.65 | 0.66 | 0.88 |
| | Winter | 0.46 | 0.47 | 0.90 |
| DEWAR LAKES | Summer | 0.60 | 0.72 | 0.91 |
| | Winter | 0.62 | 0.83 | 0.85 |
| ILIRNEJ | Summer | 0.73 | 0.76 | 0.91 |
| | Winter | 0.44 | 0.46 | 0.88 |
| KANGERLUSSUAQ | Summer | 0.37 | 0.38 | 0.90 |
| | Winter | 0.59 | 0.63 | 0.96 |
| LAINIO | Summer | 0.62 | 0.75 | 0.95 |
| | Winter | Not significant | Not significant | 0.95 |
| NADYM | Summer | 0.46 | 0.57 | 0.93 |
| | Winter | 0.74 | 0.67 | 0.90 |
| SONDRESTROM | Summer | 0.70 | 0.72 | 0.90 |
| | Winter | 0.60 | 0.63 | 0.96 |
| SUHANA | Summer | 0.84 | 0.77 | 0.93 |
| | Winter | Not significant | Not significant | 0.83 |
| SVETLOLBOVO | Summer | 0.47 | 0.37 | 0.79 |
| | Winter | 0.67 | 0.75 | 0.92 |
| WISEMAN | Summer | 0.60 | 0.64 | 0.75 |
| | Winter | 0.55 | 0.59 | 0.64 |
| WRIGLEY A | Summer | Not significant | 0.55 | 0.57 |
| | Winter | Not significant | Not significant | 0.58 |





**Table 10.** Trends of the anomaly differences for winter and summer. Trends in bold-italic are statistically significant.

| Station | | Trend [K/dec.] LST - Tair max (p-value) | Trend [K/dec.] LST- T2M (p-value) |
|---|---|---|---|
| BAKER LAKE A | Summer | -0.57 (0.30) | ***-0.80 (0.043)*** |
| | Winter | -2.04 (0.10) | -1.61 (0.28) |
| DEWAR LAKES | Summer | 0.96 (0.23) | 0.81 (0.22) |
| | Winter | -0.00 (1.0) | 0.39 (0.45) |
| ILIRNEJ | Summer | 0.06 (0.71) | 0.21 (0.44) |
| | Winter | -0.43 (0.75) | -0.33 (0.54) |
| KANGERLUSSUAQ | Summer | ***0.83 (0.008)*** | ***1.06 (0.001)*** |
| | Winter | -0.05 (0.93) | 0.17 (0.85) |
| LAINIO | Summer | 0.58 (0.08) | 0.51 (0.10) |
| | Winter | - | - |
| NADYM | Summer | ***1.56 (0.002)*** | ***1.41 (0.001)*** |
| | Winter | -0.97 (0.59) | -0.89 (0.53) |
| SONDRESTROM | Summer | -0.57 (0.11) | -0.46 (0.21) |
| | Winter | -1.13 (0.19) | -1.22 (0.16) |
| SUHANA | Summer | ***0.47 (0.028)*** | ***0.71 (0.0001)*** |
| | Winter | ***1.43 (0.01)*** | 0.80 (0.10) |
| SVEAGRUVA | Summer | -0.53 (0.45) | -0.42 (0.40) |
| | Winter | - | - |
| SVETLOLBOVO | Summer | ***1.22 (0.009)*** | ***1.14 (0.01)*** |
| | Winter | 0.43 (0.41) | 0.38 (0.55) |
| WISEMAN | Summer | 0.34 (0.62) | 0.34 (0.18) |
| | Winter | -1.39 (0.44) | -0.71 (0.08) |
| WRIGLEY A | Summer | 0.54 (0.23) | 0.17 (0.29) |
| | Winter | -1.33 (0.39) | -0.81 (0.46) |





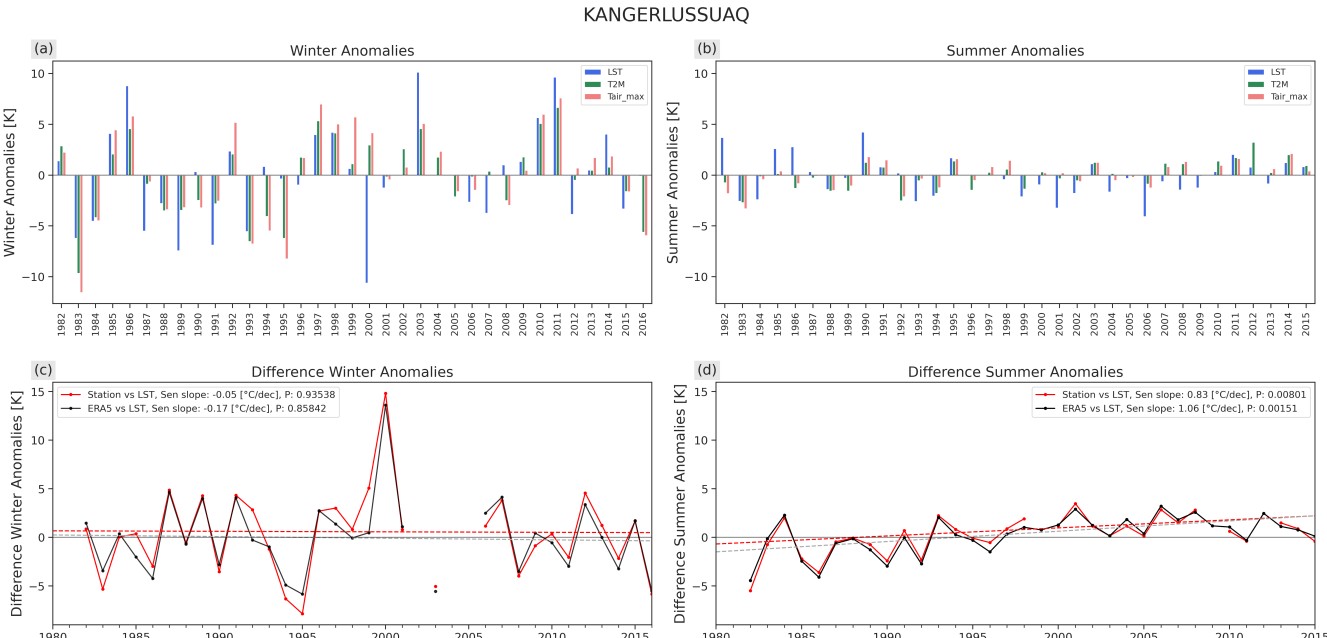

**Figure 7.** Winter and summer anomalies and the difference between anomalies of LST, T2M and Tair time series for KANGERLUSSUAQ (Greenland).

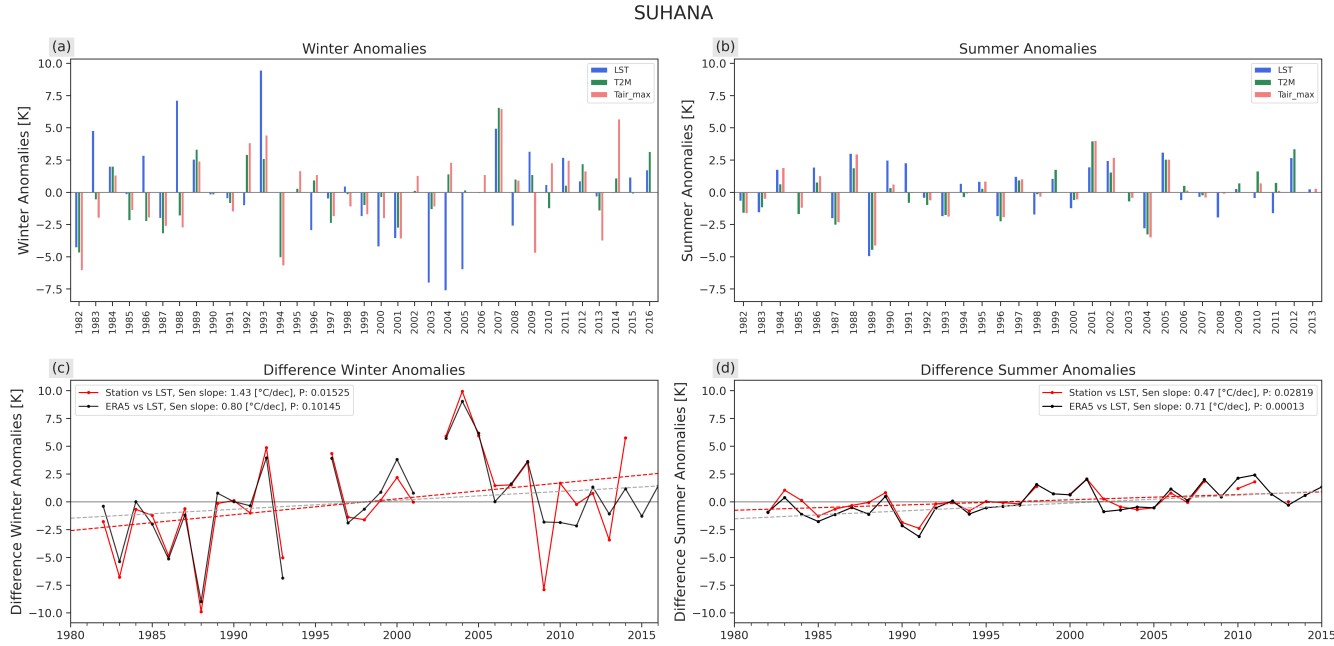

**Figure 8.** Winter and summer anomalies and the difference between anomalies of LST, T2M and Tair time series for SUHANA (Siberia).



## 4.4  LST analysis for the Pan-Arctic region

The previous sections identified a few stability issues in the current LST dataset, particularly during summer months (Table 7
and Table 10). However, these limitations are not unique to LST, as analyses of the difference between both air temperature
datasets also revealed instabilities (Table 7). While this highlights the stability and accuracy of the LST dataset, trend analysis
should be interpreted with care. The summer months are more prone to instability in the analysis than the winter months.
Therefore, the trends of the winter and summer months are computed separately for the entire Pan-Arctic region (Fig. 9). This
allows us to compare and analyze temperature changes across different seasons. Additionally, mean LST values for summer
and winter for the Pan-Arctic are calculated (Fig. 10) to further understand temperature distributions during different seasons.

Cold glaciers and mountain zones in West Canada and Alaska are well captured in the summer mean temperatures (Fig. 10).
These regions also exhibit a pronounced warming during summer and winter (Fig. 9). Warmer mean winter temperatures are
present along the Lena River in Siberia, which is also visible in the Mean Annual Ground Temperature (MAGT) map from
Obu et al. (2019). Generally, summer and winter mean LST values follow the temperature pattern shown in the MAGT map,
e.g. valleys in Russia present in the MAGT map are also visible in the mean LST values (Obu et al., 2019). The mountain range
of Yakutiya (far northeastern Russia) presents a pronounced winter warming trend. During summer that area does not show a
significant temperature trend.

During the winter period, pronounced warming can also be observed in the south of Greenland as well as in eastern Canada
(Fig. 9c). Vandecrux et al. (2024) analysed firn and ice temperature at 10m below the surface (T10m) across the Greenland ice
sheet and found a general warming trend across the ice sheet. Parts of south Siberia show negative winter and summer trends.
Similar cooling trends for winter are also visible in the AVHRR Polar Pathfinder product over Siberia, as well as warming
trends along the Siberian north coast during winter (Key et al., 2016). Large areas at latitudes > 70° suffer from persistent
cloud cover and are not covered by satellite LST data. Summer LST trends reveal warming in the north Siberian lowlands and
the Lena Delta area. Northern Canada and most of Greenland also show warming. Cooling trends are visible in the summer
period in the south of Siberia and southern Canada.





**Figure 9.** (a) and (b) show the trends for the winter period and (c) and (d) the trends for the summer period. The left panels ((a) and (c)), show the trends independent of their significance. The right panels (b) and (d) show only the significant trends, i.e. areas with statistically insignificant trends (p>0.1) are masked out.





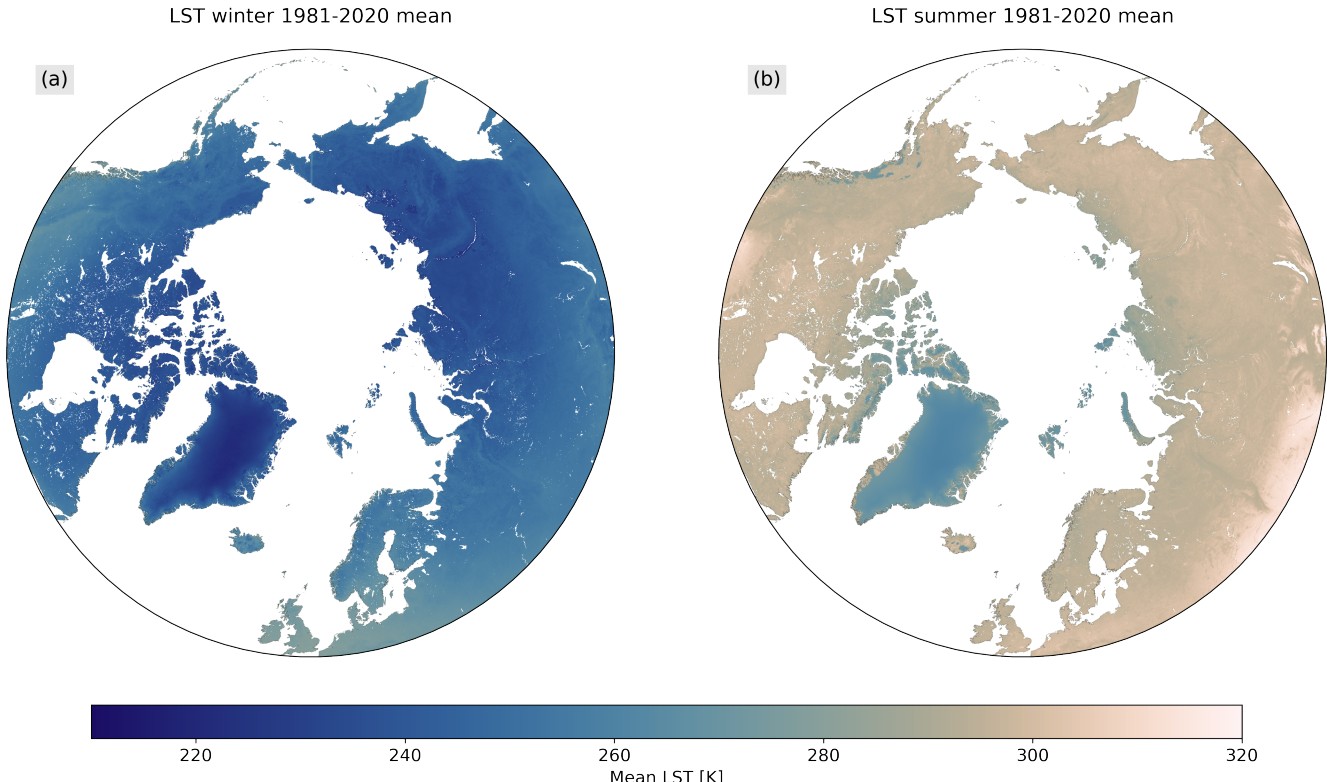

**Figure 10.** Mean LST (1981-2020) for (a) winter and (b) summer.

## 5 Discussion

### 5.1 Datasets

#### 5.1.1 EUMETSAT AVHRR FDR

The AVHRR data used in this study are not considered a fundamental climate data record (FCDR) because the calibration for
the thermal channels is satellite-specific, and uncertainty quantification is missing. The absence of thermal inter-calibration
of the IR sensors onboard AVHRR might introduce sensor discontinuities in the dataset (EUMETSAT, 2023c). Therefore, the
stability of this dataset needs to be thoroughly assessed before using it for climatological analyses. Solutions to fix this issue
might be implemented in the next release of the FDR. However, in the present study, sensor discontinuities were not detected.
AVHRR data are affected by orbital drift and, although this effect is negligible for the northern hemisphere and monthly means,
this still might introduce artificial trends. Different solutions to correct for orbital drift exist (Ma et al., 2020; Julien and Sobrino,
2022). However, for the moment, none of the existing LST datasets applies such a correction to the entire AVHRR time period.





The present study revealed artificial trends during summer months for a few stations when compared to ERA5-Land T2M and in situ air temperature.

### 5.1.2 ERA5-Land T2M

T2M data from reanalyses have the advantage that they are continuous and free of data gaps. However, the coarse resolution of ERA5-Land (9 km), represents a challenge when compared with variables such as LST, which have high spatial variability and are linked to intrinsic properties of the surface such as roughness and moisture (Hulley et al., 2014). Some discrepancies between ERA5-Land T2M and the EUSTACE station Tair data are visible in the anomaly analysis (Fig. 7 and Fig. 8). Previous work performed for the Chinese Qilian Mountains (Zhao and He, 2022) found an average RMSE of 2.2 °C between ERA5-

Land and air temperature measurements from weather stations, which was mainly attributed to elevation differences between both data sources. However, ERA5-Land performed well in estimating trends. Another study from the northeast of Brazil (de Araújo et al., 2022), which also compares air temperature data from stations to ERA5-Land, indicates that ERA5-Land generally underestimated average air temperature values.

### 5.1.3 EUSTACE Tair

EUSTACE air temperature data are homogenised and have undergone break detection and quality checks (Brugnara et al., 2019). Large parts of East Siberia, northern Canada and Alaska only have a few EUSTACE weather stations on their territory (Rayner et al., 2020). In general, stations are placed close to settlements and road access, meaning it is difficult to obtain quality weather station data from remote places. Satellite imagery represents, therefore, a valuable source for obtaining temperature data from remote places. In the current study, many stations were discarded due to their proximity to large water bodies or

the coast, which would impact the comparison with remote sensing data. In Greenland, two EUSTACE stations are located in close vicinity from each other (KANGERLUSSUAQ and SONDRESTROM). However, they present very different results when compared to ERA5-Land data and satellite data (see Tables 7 and 9). Both stations are assigned to different ERA5-Land and satellite pixels. This area in Greenland is composed of deep fjords with steep hillslopes, which might not be captured well by the spatially coarse model data and satellite data.

### 5.1.4 Pan-Arctic AVHRR LST


The AVHRR LST dataset developed in this study covers 40 years and represents a valuable data source, complementing data from models and weather stations for obtaining temperature information at a hemispheric scale. A dynamic snow and vegetation cover mask is integrated into the LST algorithm to assign correct emissivities to snow-covered pixels. This is particularly important for cryospheric research at high latitudes. Most of the existing LST datasets do not use a dynamic snow mask and

assign snow emissivity based on a static LSE map (Ma et al., 2020). Snow cover onset and snow melt onset are events that can be captured by satellite thermal imagery and are of particular interest for the thermal regime of the ground (Grünberg et al., 2020; Hammar et al., 2023). Westermann et al. (2012) highlighted the importance of using an accurate cloud mask when using




LST data for permafrost modelling, as this can lead to high uncertainties and a lack of accuracy. The Pan-Arctic AVHRR LST dataset incorporates the latest cloud mask from the CLARA-A3 database (Karlsson et al., 2023b) and a low cloud probability

threshold (see Sect. 3) was used to avoid cloud contamination.

LST trends computed at different stations (Table 8) are positive and lie in the same ranges as trends computed for Tair and T2M. Furthermore, LST temperature trend maps (Fig. 9) highlight areas that are particularly sensitive to arctic amplification and present pronounced warming trends. LST trends computed for the summer and winter periods at a hemispheric scale reveal distinct warming areas but also some regions with cooling trends. Maturilli et al. (2019) determined air temperature trends from

weather station data in Ny-Ålesund (Svalbard) from 1993 to 2017 and found that the strongest warming trend occurred during the winter season. They found a summer warming trend of +0.6K/decade, which corresponds to the LST warming trend for Svalbard shown in Fig. 9. Compared to air temperature trends, satellite-derived LST trends can present a cold bias as only clear-sky days are considered in the LST generation (Westermann et al., 2017). Therefore, the all-sky LST trends might be even higher. All-weather LST datasets have been generated in the past by combining energy balance modelling or reanalysis

LST data with LST TIR data (Martins et al., 2019; Zhang et al., 2021).

## 5.2 Point to pixel comparison

The landscape in the Pan-Arctic region is very heterogeneous, and e.g. complex wetland systems are difficult to map from satellite imagery (Olefeldt et al., 2021). Land cover data such as from the CCI project are thus prone to high uncertainties in the high northern latitudes. The nominal spatial resolution of AVHRR GAC pixel is 4 km. Air temperature measured at

a nearby location might differ considerably from the corresponding AVHRR LST and depends on vegetation type and water content of the ground.

## 5.3 Comparison of LST, Tair and T2M

Day length and, consequently, solar irradiance cover a wide range in the Arctic. During winter, there is constant night, and Tair is in close agreement with LST (Hachem et al., 2012; Urban et al., 2013). Furthermore, during winter, most of the Pan-Arctic

region is covered by snow. Snow cover variations directly influence LST (Thiebault and Young, 2020). LST and air temperature anomalies exhibit a strong correlation (mean (r) > 0.6), and LST anomalies show a similar pattern to air temperature anomalies (Fig. 7 and 8). LST anomalies present a greater magnitude, which can be explained by a greater amplitude in the LST diurnal cycle than in the air temperature diurnal cycle. Differences in winter anomalies are more pronounced than summer anomalies. This can be explained by the clear-sky bias that occurs in satellite LST data (Westermann et al., 2012): cloud cover affects

the winter period more than the summer period. Such a cooling bias was also observed over the Greenland Ice sheet over ice surface temperature (Hall et al., 2012). Regarding monthly means, the air temperature mean is higher than the LST mean value, which only considers clear sky days. Under cloudy skies, air and surface temperature are in close agreement (Obu et al., 2019). A few missing data occurred in the EUSTACE, which made the ERA5-Land data more reliable for such an analysis. However, EUSTACE records two temperatures per day, Tmax and Tmin. Therefore, for analysis at daily frequency, EUSTACE

might be more suitable, especially for satellites having an overpass time close to noon. Winter data proved to be stable over all



investigated stations. This makes this dataset particularly interesting for studying the winter months in the high latitudes. The winter period is a particularly active time for the ecosystem in the Arctic (Berge et al., 2015), but it tends to be understudied.

Finally, trends for the three different datasets computed at stations with good temporal stability (Table 8) showed good agreement, with LST trends generally exhibiting stronger positive trends than air temperature. In addition, LST winter trends for mid-Siberia (Fig. 9) showed similar values as in Waring et al. (2023). The regions showing positive trends in LST are generally associated with lichens, moss and herbaceous land cover, whereas regions with negative trends are mostly located in forested areas. Mildrexler et al. (2011) revealed the cooling effect of forests on LST, which might slow the general temperature increase in the Arctic. The presence of thermokarst lakes and wetlands as depicted in Olefeldt et al. (2021) might also slow the warming trends slightly. The Yamal peninsula, for example, contains many water bodies and thermokarst lakes and shows a slower warming trend than an area with a majority of barren soil. Hemispheric LST data can also be used to highlight fast-warming areas that might lead to abrupt permafrost thaw, which in turn influences carbon fluxes (Treat et al., 2024). LST also has the advantage over air temperature to be more sensitive to changes in vegetation density (Mildrexler et al., 2011), which makes LST a particularly interesting variable for cryospheric research.

## 6  Conclusions

This study presents the workflow to derive a Pan-Arctic LST dataset from the EUMETSAT AVHRR GAC FDR and validates the new LST product against in situ LST. AVHRR LST is derived with the generalized split-window algorithm (Wan and Dozier, 1996) and the corresponding RTM is performed with RTTOV v.13, based on a new calibration database (Ermida and Trigo, 2022). This ensures an optimal representation of atmospheric conditions in the Pan-Arctic region. This LST dataset utilises a recent cloud mask with notable improvements compared to previous cloud products (Karlsson et al., 2023b), which is of particular importance for LST retrieval. The Pan-Arctic AVHRR LST product showed good performance, and validation results lie in the range of similar products. The new LST product is assessed for stability in the Pan-Arctic region by comparing it to air temperature data from weather stations and T2M data from ERA5-Land. Twelve weather stations belonging to the EUSATCE global station dataset are chosen based on several criteria: latitude > 50°N, a minimum of 30 years of overlapping data, and homogeneous land cover over at least one GAC pixel. LST trends and variability are compared to ERA5-Land T2M and EUSTACE Tair max data. The correlation coefficients between the datasets indicate good agreement, (r) > 0.9 for the monthly mean correlation and ~0.5 to ~0.8 for the anomaly analysis. The analysis of the differences in anomalies showed slightly significant trends for the summer months but no artificial trends in the data during the winter months when solar irradiance is absent. LST trends for the winter and summer periods were computed for the entire Pan-Arctic region and revealed spatially varying trends. In winter positive trends in the south of Greenland, Siberia and eastern Canada were revealed. Summer temperature trends highlight the fast warming of the Greenland Ice sheet and a large region in the vicinity of the Lena Delta. Its good accuracy and emissivity retrieval based on dynamical vegetation and snow masks make this product suitable for a wide range of research applications in the Pan-Arctic region. However, future research investigate a possible spatial upsampling of





the dataset, e.g. to 1 km spatial resolution. Orbital drift correction with a robust method would allow to extend the study to lower latitudes.

520 **Appendix A:  Appendix A**

**A1    Anomaly differences**

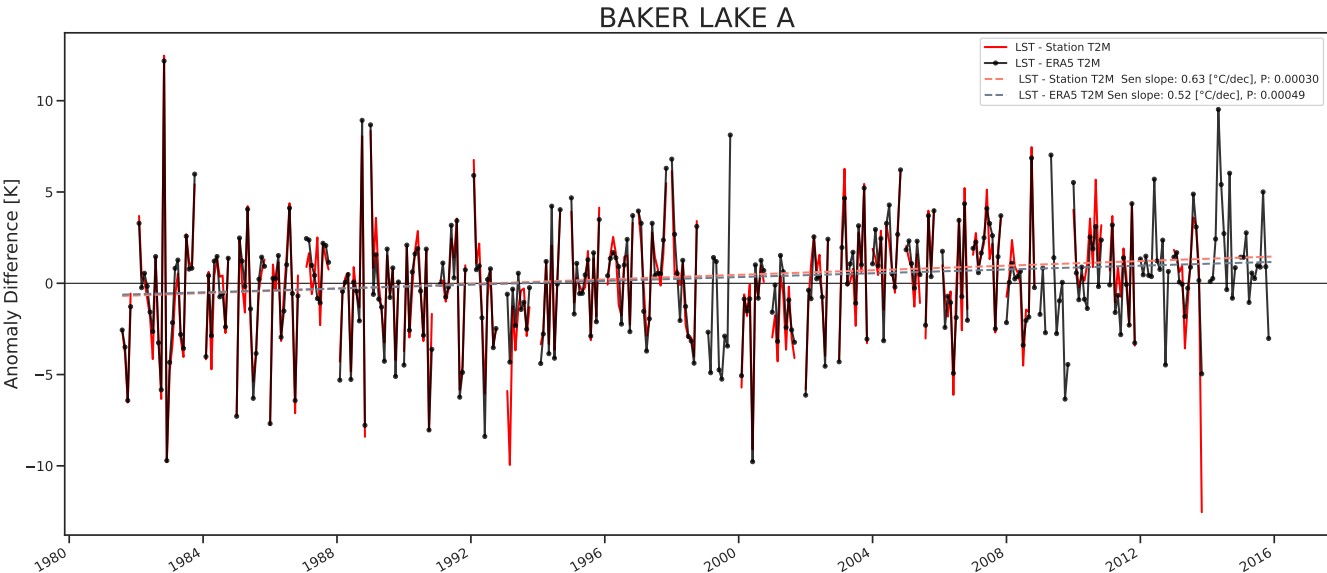

**Figure A1.** Differences of the anomalies at BAKER LAKE A (Canada) as a time series.



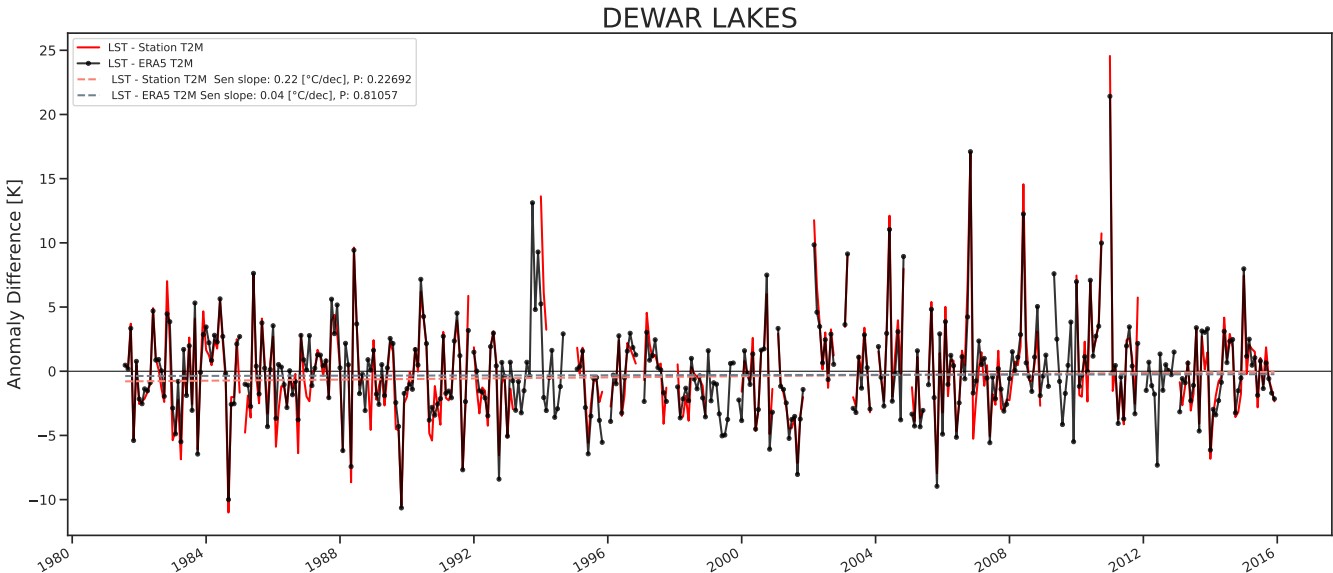

**Figure A2.** Differences of the anomalies at DEWAR LAKES (Canada) as a time series.

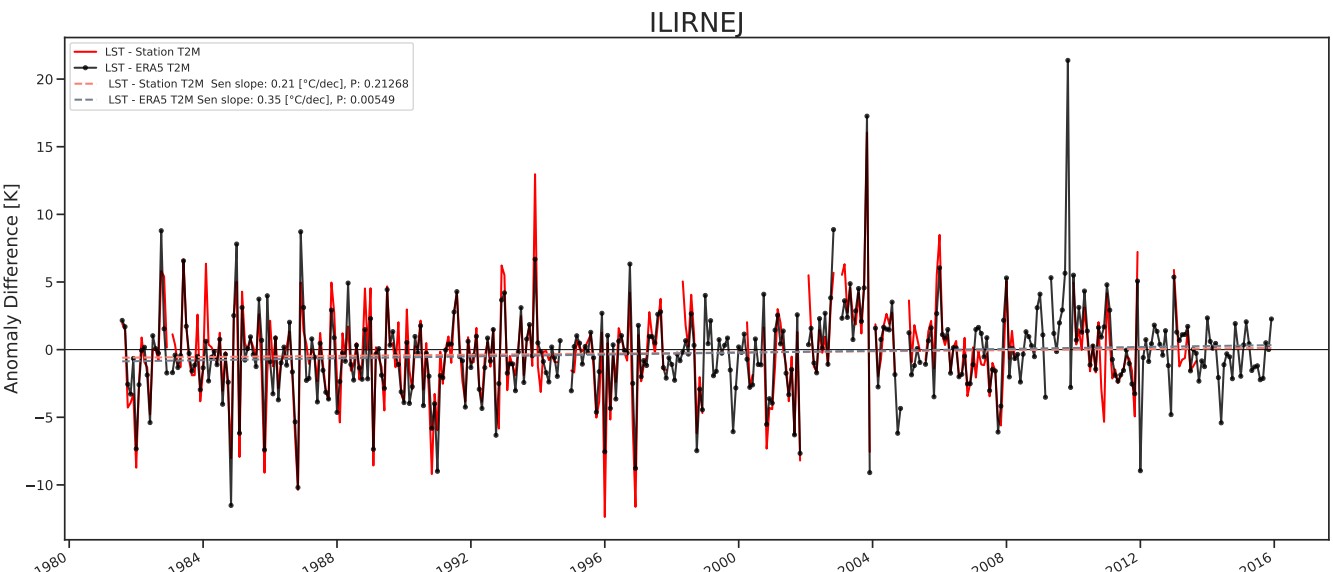

**Figure A3.** Differences of the anomalies at ILIRNEJ (East Siberia) as a time series.



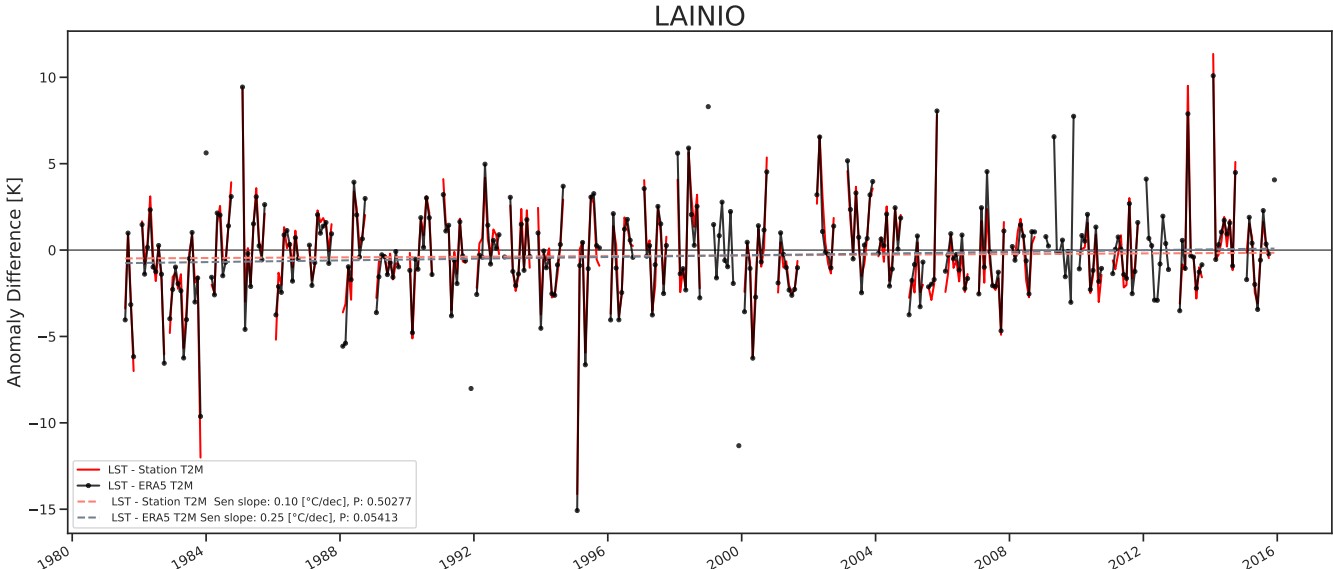

**Figure A4.** Differences of the anomalies at LAINIO (Norway) as a time series.

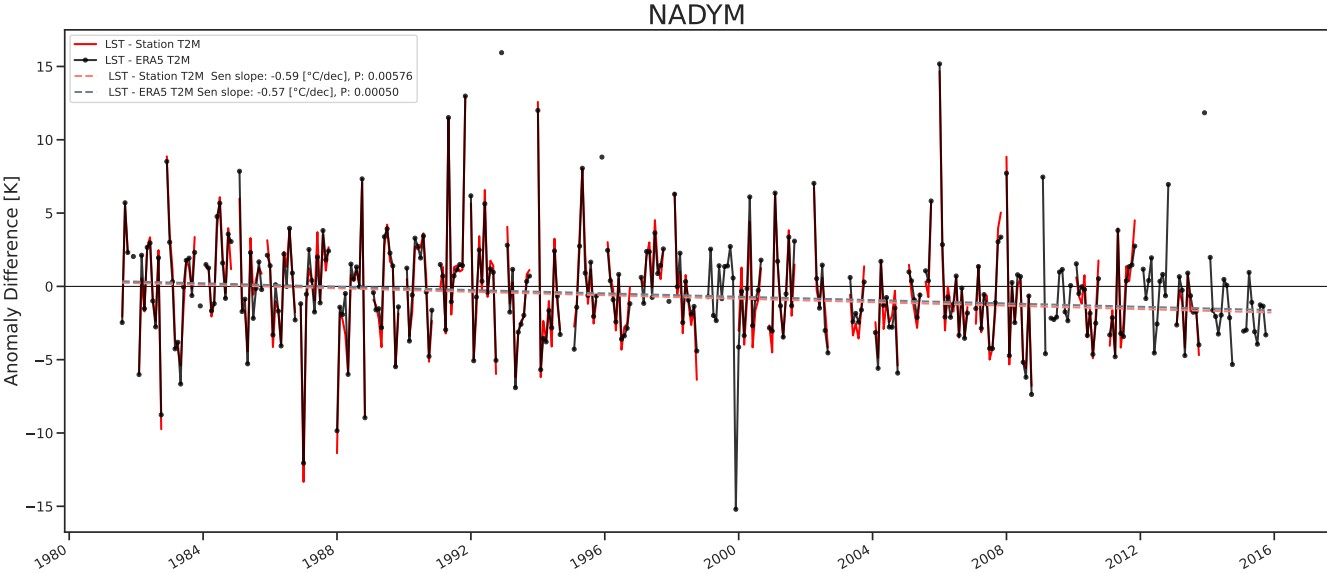

**Figure A5.** Differences of the anomalies at NADYM (Siberia) as a time series.



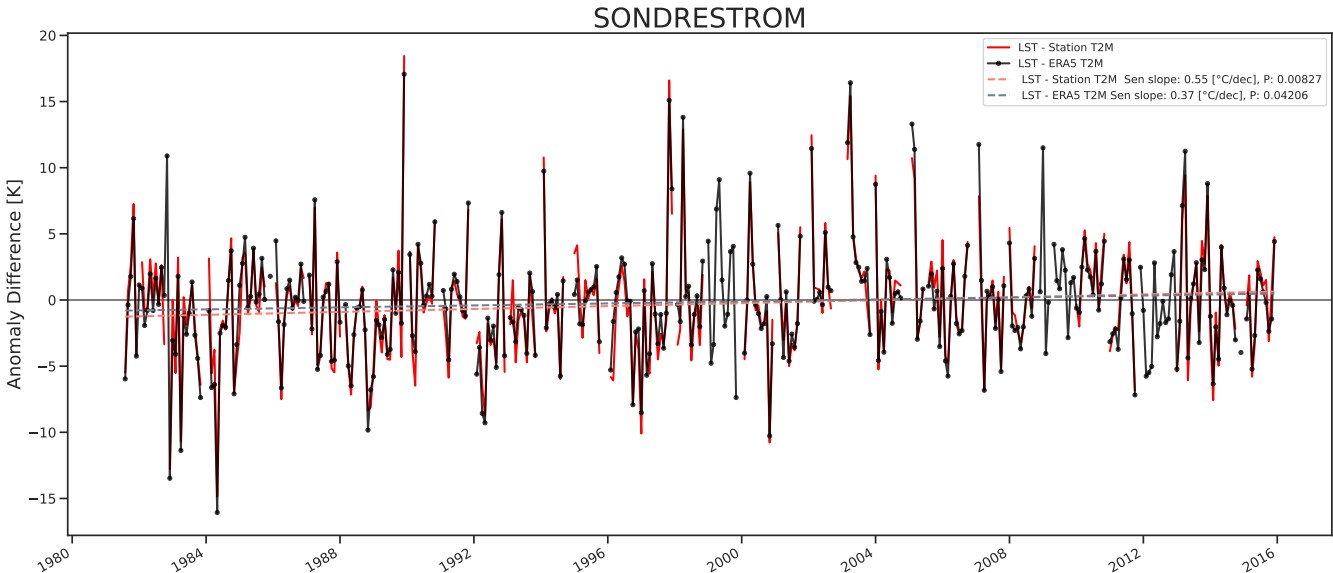

**Figure A6.** Differences of the anomalies at SONDRESTROM (Greenland) as a time series.

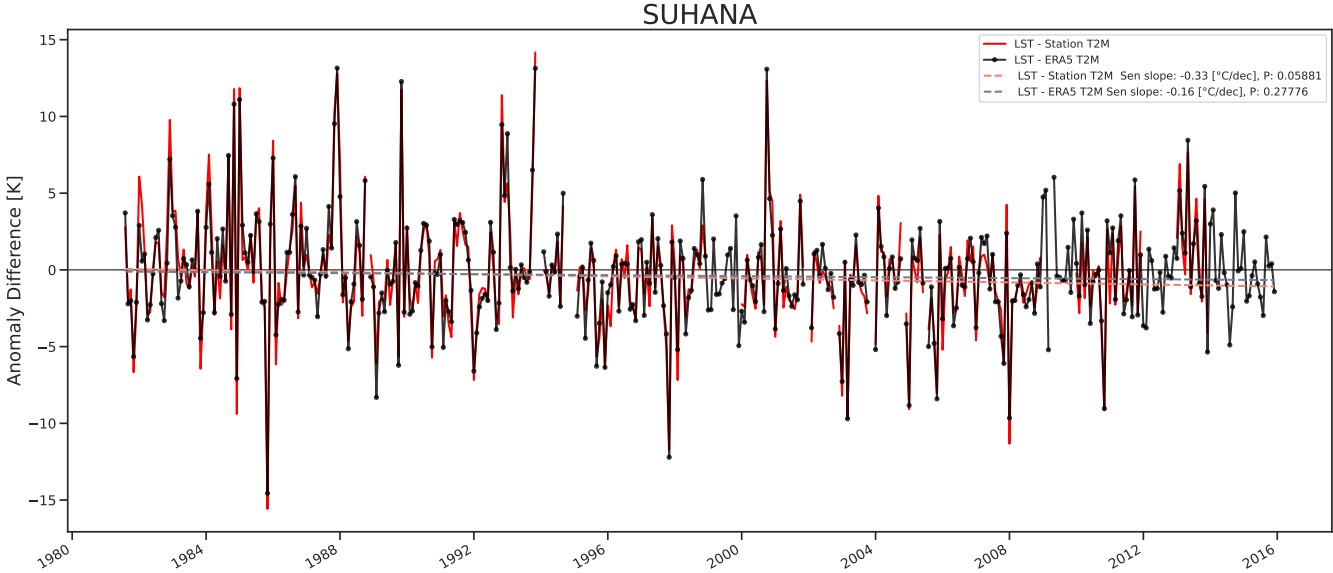

**Figure A7.** Differences of the anomalies at SUHANA (Siberia) as a time series.





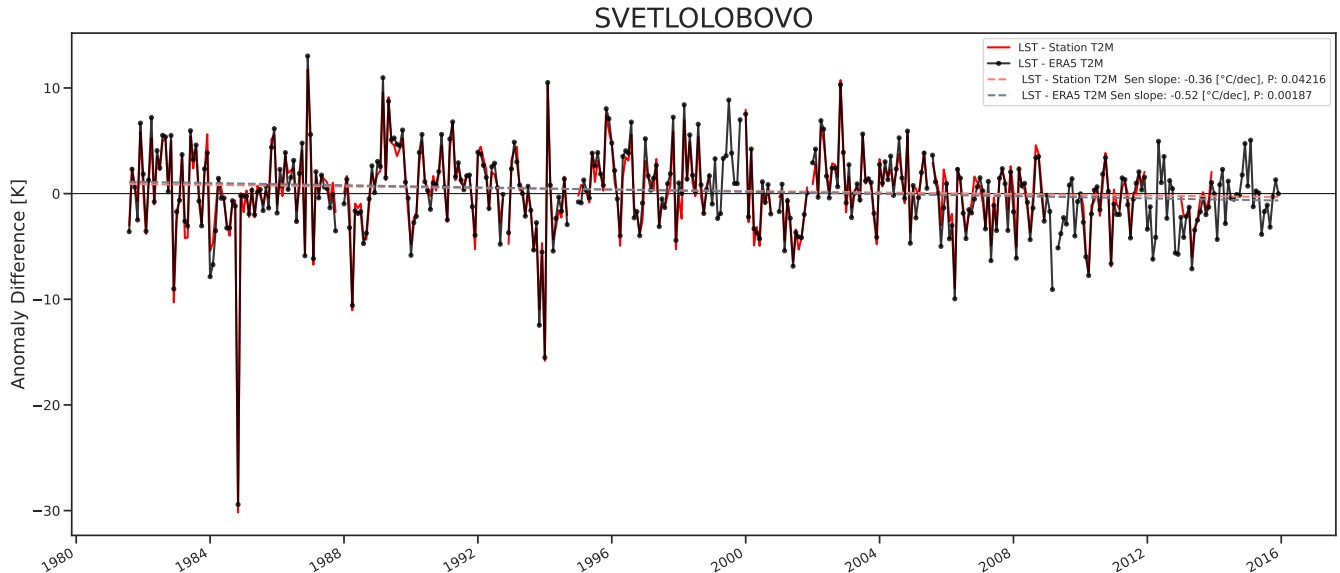

**Figure A8.** Differences of the anomalies at SVETLOLBOVO (Siberia) as a time series.

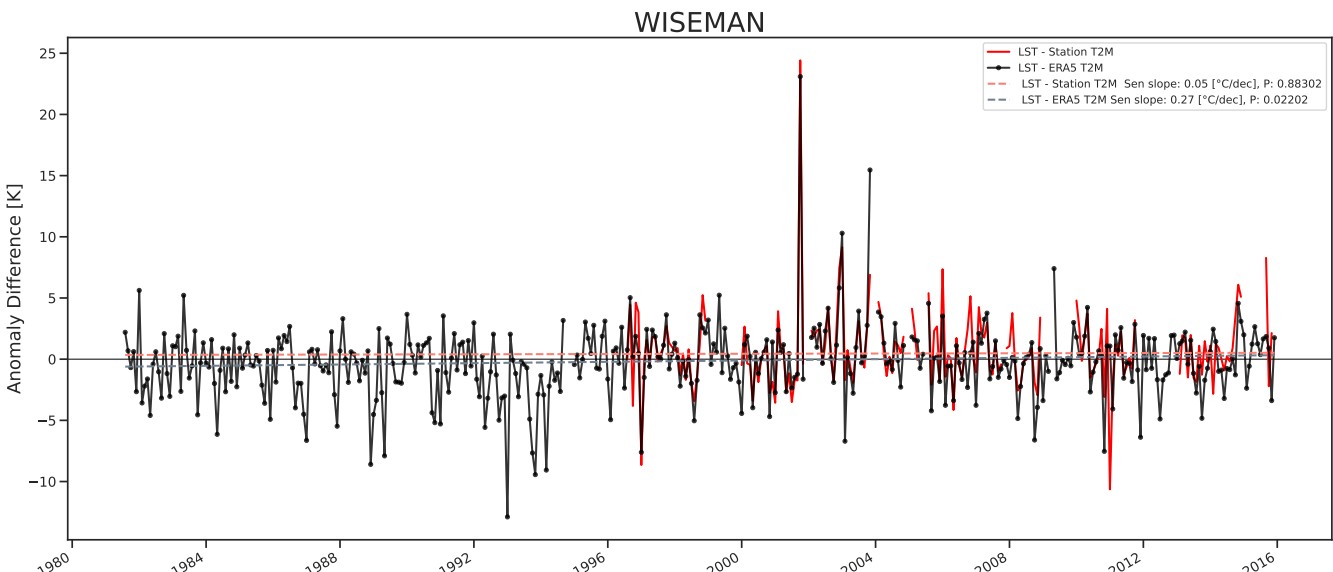

**Figure A9.** Differences of the anomalies at WISEMAN (Canada) as a time series.



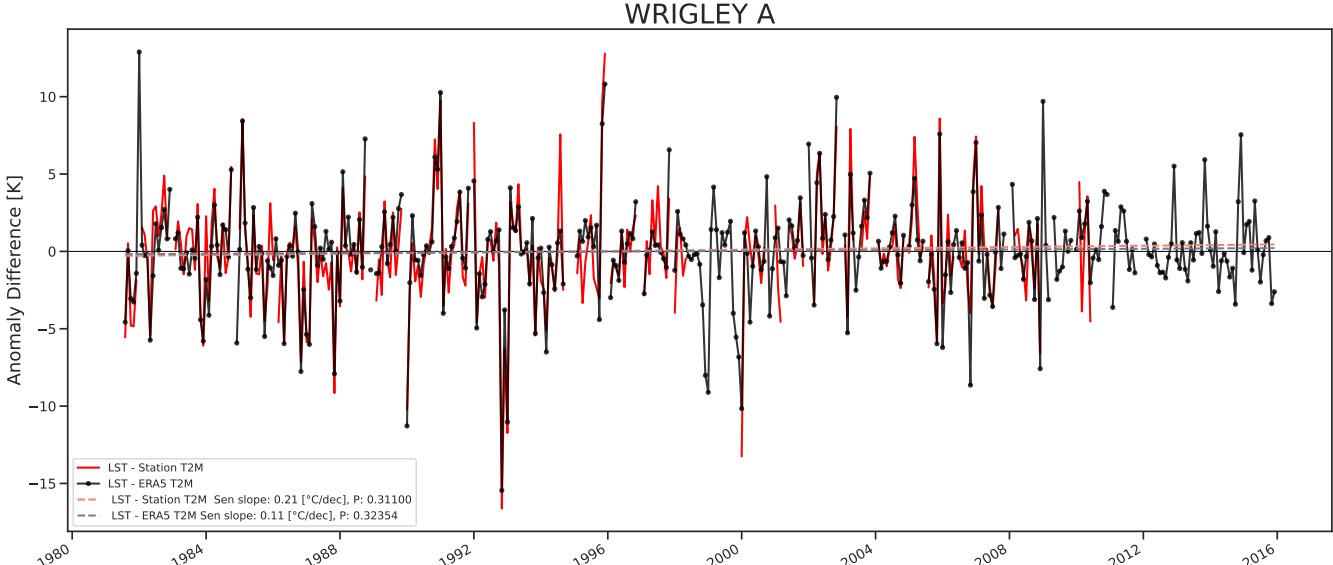

**Figure A10.** Differences of the anomalies at WRIGLEY A (Alaska) as a time series.

*Data availability.* The EUMETSAT AVHRR FDR, the basis for this study, is freely available through the EUMETSAT Data Portal: http://doi.org/10.15770/EUM_SEC_CLM_0060. In addition, the CCI + Snow project datasets are available here: https://climate.esa.int/en/odp/#/project/snow, and the clear-sky database is accessible through Zenodo (https://doi.org/10.5281/zenodo.5779543).

525 *Author contributions.* The idea was conceptualized by SD and refined by SW. The software development, data processing, data analysis and writing were mainly conducted by SD. FMG and SW supervised the manuscript writing and provided supervision of the research.

*Competing interests.* The authors declare that they have no conflict of interest.

*Acknowledgements.* We thank the Dr. Alfred Bretscher Stipendium for climate and air pollution research from the University of Bern for funding this project. We also acknowledge the team at EUMETSAT that compiled the AVHRR FDR data record. We acknowledge the use of
530 the scientific color maps developed by Fabio Crameri (https://doi.org/10.5281/zenodo.1243862). We also want to mention the open-source Python library Xarray, which has been a great ressource for this project.





*Financial support.* This work was supported by the Dr. Alfred Bretscher Stipendium for climate and air pollution research from the University of Bern.



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
