# Peer review of "Temporal stability of a new 40-year daily AVHRR Land Surface Temperature dataset for the Pan-Arctic region"

_EGUsphere, 2024_

## Author Comment (AC1)

We thank Referee #1 for the useful and detailed comments. We have followed the suggestions and modified our manuscript as described in the line-by-line answers below. Note that the reviewer's comments are in black text, our answers in red and the changes in the manuscript are indicated in red italic.

Specific comments

1. In [Page 7, Table1], the unit of latitude, longitude, and elevation in the table head should be indicated

The units were added to the table head.

2. The 2m-air temperature is derived from the ERA5-Land product, but skin temperature and total column water vapor are derived from MERRA-2. I would like to know why different reanalysis data are used, as ERA5 seems to provide these parameters as well, with a higher spatial resolution.

Two datasets were chosen for this study: MERRA-2 and ERA5-Land.

MERRA-2 assimilates space-based observations of aerosols and accounts for ice sheets, ensuring accurate data for regions like Greenland. The GLASS product (Ma et al., 2020) has been generated with MERRA-2 products: using MERRA-2 for our study simplifies comparisons between the GLASS and our product. Our study uses skin temperature and total column water vapor from MERRA-2 to determine the atmospheric conditions at each pixel and to select the split-window (SW) coefficients from the look-up table.

In contrast, air temperature data from ERA5-Land are used for the correlation and stability analysis. Using separate sources of reanalysis data helps keeping the stability analysis independent from the SW coefficient assignment process. Furthermore, ERA5-Land has been fully validated in the Arctic region in a previous study on climate indices for the northern high-latitudes regions (Rantanen et al., 2023). Finally, in the stability analysis we compared point data (weather station data), satellite data and a reanalysis product: in order to optimize the spatial representativeness in the point-to-pixel comparisons, the reanalysis product with the highest spatial resolution has been chosen.

The above choices are now explained in a corresponding paragraph added to subsection *#2.4 Auxiliary data*.

**2.4 Auxiliary data**

*Skin temperature (Tskin) and Total Column Water Vapor (TCWV) from the MERRA-2 reanalysis dataset (M2T1NXSLV, variables are labelled TS and TQV). The data come at hourly temporal resolution with a spatial resolution of 0.5° x 0.625°. Nearest neighbour resampling*

*was performed to match the AVHRR spatial resolution and scanline time, i.e. as in the work of Ma et al. (2020). MERRA-2 is preferred over other reanalysis products with finer spatial resolution to allow comparison with the GLASS product (Ma et al. 2020) and to keep the LST retrieval independent from ERA5-Land, which will be used for the stability analysis.*

3. The accuracy validation of LST estimations is conducted over SURFRAD and KIT sites, while the application and analysis focus on the Pan-Arctic region. If there are available sites in the Arctic region to provide a more intuitive validation?

Data from the Atmospheric Radiation Measurement Climate Research Facility *US* Department of Energy (ARM) site at the North Slope of Alaska (NSA) are available from 2007 to 2012. These data have previously been used in the ESA GlobTemperature project and undergone quality control procedures. The data from the NSA site in northern Alaska has been integrated into our validation and the corresponding figure (Fig. 4) and table (Table 2) updated accordingly. The surface is very heterogeneous at the NSA site, the station being close to lagoons (North Salt Lagoon and Imikpuk Lake), and very close to the coast. This explains why the performances are much worse during summertime than during wintertime when the entire area is snow and ice covered.
In our work we rely exclusively on data that have been quality controlled and previously used to validate global operational LST datasets, e.g., the ESA GlobTemperature and LST_CCI datasets.

[Figure]

*Figure 4. AVHRR LST versus in situ LST at (a) Bondville (BND), (b) Desert Rock (DRA), (c) Fort Peck (FPK), (d) Goodwin Creek (GCM),(e) Penn. State Univ (PSU), (f) Sioux Falls (SFA), (g) Southern Great Plains (SGP), (h) Table Mountain (TBL), (i) Evora (EVO), (j) Lake Constance (BOD), (k) North Slope of Alaska (NSA). Red represents daytime measurements and blue represents nighttime measurements. Match-up periods are provided in the text.*

4. In [Page 8], the authors mention that "The Copernicus digital elevation model (DEM) GLO-90 upscaled to 0.05° spatial resolution is used for the RT modelling". The geopotential

height has been included in the used atmospheric profile dataset which can be used to calculate elevation, why still using additional data sources?

Geopotential height was used for the generation of the database (Ermida et al., 2022) but was not saved in the final disseminated product (personal communication from Dr. Ermida).

5. In [Page 12], the authors mention that "Pixels that have a cloud fraction higher than 0.1 are removed, and the average of the remaining pixels is computed". As far as I am concerned, in order to eliminate potential cloud contamination, LST averaging should be performed only when all pixels within the window are clear.

Thank you for pointing this out. Our work uses a probabilistic cloud mask (named CMAPROB), part of the level-2b product of CLARA-A3 (CM-SAF, CLARA-A3 Product User Guide, 2023)., i.e., not a cloud fraction layer. This was a misrepresentation in the manuscript and has been corrected accordingly. Pixels with a cloud probability below 0.1 are considered clear. This is a compromise between data availability and avoiding cloud contamination; particularly over snow and ice surfaces, this is a reasonable assumption. One sentence explaining the above has been added to the subsection _#3.4 LST AVHRR time series generation_.

**3.4 LST AVHRR time series generation**

_Depending on the heterogeneity of the land cover, between four and nine AVHRR LST GAC pixels are extracted around each station. Pixels that have a_ cloud probability _higher than 0.1 are removed, and the average of the remaining pixels is computed._ This cloud probability threshold is a compromise between data availability and avoiding cloud contamination.

6. In [Page 13, line 265], the explanations of symbols in equation (4) to (6) are missed.

Symbol explanations were added to equations (4) to (6) on [Page 13].

7. In [Page 14, line 295], only nighttime observations from EVO site are used to bypass the directional effects. While for several SURFRAD sites covered by vegetation, possible directional effects may also exist, but why both daytime and nighttime data are used?

The main end-members at Evora LST validation site are evergreen trees (mainly cork oak trees) and grass. A tree crown cover (TCC) of 33% was determined from satellite data (Guillevic et al., 2013; Ermida et al., 2014). Structured canopies such as the ones in EVORA can show pronounced directional effects which influence LST estimates. These effects depend on illumination and viewing geometries (Rasmusen et al. 2011). At daytime, the EVO

site shows a high level of thermal anisotropy, with temperature differences between the tree crowns and the (dry) grass frequently exceeding 20K.

The SURFRAD station at DRA (desert Rock, Nevada) also exhibits an arid soil with small bushes leading to anisotropy, although not as pronounced as EVORA. The validation plot (Fig. 4, see comment #3) has been modified and now separates day and night information.

More details about EVO station and on anisotropy effects have been added to subsection to subsection *#4.1.2 Validation with in situ LST.*

**4.1.2 Validation with in situ LST**

*EVO is located in an evergreen oak woodland with approximately 33% of tree crown cover, which can affect the satellite-retrieved LST due to directional effects (Rasmussen et al. 2011; Guillevic et al., 2013; Ermida et al., 2014). Due to this anisotropy, the surface in EVO presents high temperature differences between trees and ground. The nighttime in situ measurements in EVO are therefore more suited than daytime observations [removed sentence]*

8. It seems that only the overall performance is shown in Figure 4. It is recommended to add accuracy metrics for daytime and nighttime, respectively, to provide a better comparison.

Please see our answer to comment #3 and #7; plots showing the day/night validation results separately were added to Figure 4.

9. The trend analysis is conducted based on monthly averaged LST of each pixel (There are fewer descriptions about this in the manuscript, thus I guess maybe all clear-sky daytime and nighttime observations of the pixel are used to calculate). However, the averaged LSTs may be seriously affected by the frequency of cloud cover. For example, for pixel A, daytime observations account for 50% proportion, whereas in pixel B, they constitute 70%. Since daytime LST tend to be larger than nighttime LST, the averaged LST of pixel B tend to be larger than A, but that may not be the true situation. Therefore, the cloud cover may lead to the incomparability between pixels. Even for the same pixel, changes in cloud cover frequency between different months may also result in temporal incomparability. Therefore, is it possible to reduce these incomparable effects, such as ensuring a

balanced distribution between day and night LSTs to calculate averaged LST? Besides, this restriction should be briefly explained in the discussion section.

The analysis was limited to daytime data: this has now been clarified in the manuscript. Only data with a cloud probability (not fraction – see our answer to comment #5) of less than 0.1 were used, thereby strongly reducing potential cloud contamination. Subsection *#3.4 LST AVHRR time series generation* of the manuscript has been modified accordingly and now provides the correct details of the cloud filtering and which data entered the time series generation.

**3.4 LST AVHRR time series generation**

*Depending on the heterogeneity of the land cover, between four and nine AVHRR LST GAC pixels are extracted around each station. Pixels that have a cloud probability higher than 0.1 are removed, and the average of the remaining pixels is computed. Daytime data from NOAA-7, 9, 11, 14, 16, 18 and 19 (satellites with ascending (northbound) equator crossing times), as well as the entire MetOp series (satellites with descending (southbound) equator crossing times), are considered for constructing the time series. The considered period for each satellite is chosen to minimise orbital drift and avoid the outage periods (EUMETSAT, 2023d). The retained periods are listed in Table 4.*

*Once the relevant periods are extracted, outlier detection is performed based on a 10-day rolling window analysis and detected outliers are removed. Daily temperature variability is very high (Mildrexler et al., 2011), and AVHRR-derived LST time series are subject to noise, therefore, monthly means are computed from the concatenated day time series for further analysis.*

**References**

CM SAF, CLARA-A3 Product User Manual, 2023, 10.5676/EUM_SAF_C/CLARA_AVHRR/V003

Ermida, S. L., Trigo, I. F., DaCamara, C. C., Göttsche, F. M., Olesen, F. S., and Hulley, G.: Validation of remotely sensed surface temperature over an oak woodland landscape - The problem of viewing and illumination geometries, Remote Sens. Environ., 148, 16–27, https://doi.org/10.1016/j.rse.2014.03.016, 2014.

Ermida, S. and Trigo, I. A Comprehensive Clear-Sky Database for the Development of Land Surface Temperature Algorithms. Remote Sens. 2022, 14, 2329. https://doi.org/10.3390/rs14102329

Guillevic, P., Bork-Unkelbach, A., Gottsche, F. M., Hulley, G., Gastellu-Etchegorry, J. P., Olesen, F. S., and Privette, J. L.: Directional viewing effects on satellite land surface temperature products over sparse vegetation canopies-a multisensor analysis, IEEE Geosci. Remote S., 10, 1464–1468, https://doi.org/10.1109/LGRS.2013.2260319, 2013.

Ma, J., Zhou, J., Göttsche, F.-M., Liang, S., Wang, S., and Li, M.: A global long-term (1981–2000) land surface temperature product for NOAA AVHRR, Earth Syst. Sci. Data, 12, 3247–3268, https://doi.org/10.5194/essd-12-3247-2020, 2020

Rantanen, M., Kämäräinen, M., Niittynen, P. *et al.* Bioclimatic atlas of the terrestrial Arctic. *Sci Data* **10**, 40 (2023). https://doi.org/10.1038/s41597-023-01959-w

Rasmussen, M., Gottsche, F. -M., Olesen F. -S. and Sandholt I. "Directional Effects on Land Surface Temperature Estimation From Meteosat Second Generation for Savanna Landscapes," in IEEE Transactions on Geoscience and Remote Sensing, vol. 49, no. 11, pp. 4458-4468, Nov. 2011, doi: 10.1109/TGRS.2011.2144604

We thank the Referee for constructive criticism and comments which significantly improved our manuscript. In the following, we provide point-by-point replies to all issues raised. Note that the reviewer's comments are in black text, our answers in red and the changes in the manuscript are indicated in red italic.

1. It is not completely clear what is the added value of having this specific dataset derived for the artic region if the authors are simply averaging all observations within a day. There are already datasets available based on AVHRR that provide daily composites (e.g. GLASS, LSA-SAF). In my opinion, it would have been more beneficial to explore the multiple passages of the different AVHRR to try to reconstruct the diurnal cycle. That would have made the dataset more unique and more useful. Having averages of whatever observations exist in a day can create high instabilities in day-to-day variability, depending on what sensors are available and cloud coverage.

We agree with the reviewer and apologize for not making this sufficiently clear in the manuscript. The presented pan-Arctic AVHRR LST dataset does not simply average all provided LST observations within one day, which are typically two for an individual satellite. In order to generate time series from the LST observations of the different NOAA satellites, these are selected based on their overpass time (i.e., not averaged); the details are presented in Table 4. In contrast, for the *EUMETSAT* Polar System (EPS) series (MetOp-1, -2, -3), which has highly stable overpass times, the observations are averaged (see Table 4). Only the stability analysis and trend analysis are performed on monthly mean values, i.e., for each individual satellite a time series of monthly mean composites is created. Subsection *#3.4 Time series generation* now explains and clarifies the above points and the data description in subsection *#2.1 EUMETSAT AVHRR FDR* has been expanded.

The GLASS product (Zhou et al.2019; Ma et al. 2020) is based on the Long-Term Dataset Records (LTDR) (Pedelty et al., 2007) and is built on the SeeBor V5.0 (Borbas et al. 2005) data and TIGR2000 V1.2. Compared to these two profile databases, the novel calibration database from Ermida et al. (2022) is based on the recent ERA5 reanalysis and therefore exhibits high temporal and spatial coverage as well as improved good vertical resolution (137 levels). In addition, the profiles were selected with a dissimilarity criterion, ensuring

that less common atmospheric conditions are also included. Furthermore, the GLASS product has considerable data gaps above 45° latitude, which can be attributed to cloud masking (see Fig. 11 from Ma et al., 2020), and to its emissivity computation relying on visible channels, which are unavailable during the polar night. Our pan-Arctic AVHRR LST dataset utilizes a probabilistic cloud mask provided in the CLARA-A3 dataset (Karlsson et al., 2023). In addition, in our workflow, emissivity retrieval for the snow- and ice-covered areas is based on snow water equivalent (SWE) data retrieved from passive microwave radiometer (PMR), which are also available during polar night (Solberg et al. 2021).

[Figure]

**Figure 11.** Monthly averaged ODC LST retrieved from NOAA-14 data for 1999 normalized to 14:30 ST: **(a)** March, **(b)** June, **(c)** September, **(d)** December.

In order to illustrate the better availability of our pan-Arctic AVHRR LST dataset compared to the GLASS product, in section *#4.1 LST validation results* a subsection has been added that showcases the differences between the two products. Both plots below show the differences between our pan-Arctic AVHRR LST dataset (black), the GLASS product (red) and the ODC corrected GLASS product (blue). In the high northern latitudes, the GLASS product is only available during summer months. This is particularly visible for the SVEAGRUVA site (SVALBARD), where very few GLASS observations are available. Also at BAKER LAKE A our product presents considerably more and slightly higher values, which can be explained by the different cloud masking and emissivity computation.

[Figure]

Concerning the LSA SAF LST products mentioned by the Referee: the EDLST dataset from LSA-SAF is based on AVHRR-MetOp, provides uncertainty estimates and has been intensively validated. However, the EDLST time series starts with the first MetOp satellite in 2015 (https://lsa-saf.eumetsat.int/en/data/products/land-surface-temperature-and-emissivity/), while our dataset covers a 40-year period of AVHRR instruments. The other LSA-SAF LST products are based on MSG/SEVIRI data, i.e., they do not cover the high latitudes.

The introduction of the manuscript has been modified to emphasize these differences and the benefits of our Pan-Arctic AVHRR LST dataset compared to already existing datasets.

**2.1 EUMETSAT AVHRR FDR**

The FDR contains AVHRR reflectance and brightness temperatures for each available orbit and channel. The daily AVHRR data from one satellite provides nearly complete coverage of the globe. *The dataset provides for each satellite twice-daily composites (one daytime overpass and one nighttime overpass).* AVHRR GAC measurements have been processed using the PyGAC software –a Python software package to read and transform AVHRR data in GAC format- (https://pygac.readthedocs.io/en/latest/#), including the conversion from counts to reflectance or brightness temperature and cross-calibration of the visible channels of the AVHRR sensor.

**3.4 LST AVHRR time series generation**

Depending on the heterogeneity of the land cover, between four and nine AVHRR LST GAC pixels are extracted around each station. Pixels that have a cloud *probability* higher than 0.1 are removed, and the average of the remaining pixels is computed. *Daytime* data from NOAA-7, 9, 11, 14, 16, 18 and 19 (*satellites with ascending (northbound) equator crossing times*), as well as the entire MetOp series (*satellites with descending (southbound) equator crossing times*), are considered for constructing the time series. The considered period for each satellite is chosen to minimise orbital drift and avoid the outage periods (EUMETSAT, 2023d). The retained periods are listed in Table 4.

Once the relevant periods are extracted, outlier detection is performed based on a 10-day rolling window analysis and detected outliers are removed. Daily temperature variability is very high (Mildrexler et al., 2011), and AVHRR-derived LST time series are subject to noise, therefore, monthly means are computed from the *concatenated day* time series for further analysis.

**#4.1.3 Comparison with the GLASS dataset**

*The pan-Arctic AVHRR LST dataset is compared against the well-established GLASS product (Zhou et al. 2019, Ma et al. 2020), that provides twice daily LST observation for the whole globe for the 1980-2000 period. Figures 5 and 6 present a comparison of monthly means at two stations located in the Arctic (BAKER LAKE A and SVEAGRUVA). The classical GLASS LST, the orbital drift corrected (ODC) GLASS LST and the pan-Arctic AVHRR LST are compared. In the high northern latitudes, the GLASS product is only available during summer months. This is particularly visible for the SVEAGRUVA site (SVALBARD), where very few GLASS observations are available. Also at BAKER LAKE A our product presents considerably more and slightly higher values, which can be explained by the different cloud masking and emissivity computation.*

[Figure]

*Figure 5. Monthly means LST product comparisons at BAKER LAKE A.*

[Figure]

*Figure 6. Monthly means LST product comparisons at SVEGRUVA.*

2. For the same reason, I'm not convinced the dataset is appropriate for trend, and specially not for anomaly analysis. If the time of observation that goes into the average keeps changing, then there is just too much instability in the series.

Again, we agree and apologize for not describing our approach to AVHRR LST time series generation clearly enough; please refer to our in-depth answer to point #1.

Subsection *#3.4 Time series generation* has been modified and expanded to clarify the differences in generating LST time series for the NOAA and MetOp satellites. For each individual satellite the selected time period (Table 4) has been chosen to minimize the effect of orbit drift. Furthermore, winter data (December and January) are analyzed separately from the summer data to investigate the influence of the orbital drift on the trend analysis.

3. Also, in terms of algorithm calibration, here there was a unique opportunity to explore an algorithm more suited for the specific conditions of the Artic. That maybe would allow using a higher range of view angles, resulting in an even larger sampling of observations through the day. The same in terms of the calibration database, why not tailor the database to the more specific conditions of the Artic? Using a generic algorithm and database that are valid over the whole globe is something that is already available in other products.

The Generalized Split Window (GSW) algorithm we have employed (Wan & Dozier, 1996) is well-established and used for operational LST products (e.g., LST products from LSA-SAF). This algorithm is optimal for sensors with two TIR channels centered at 11 and 12 μm, which is the case of AVHRR. The GSW algorithm was compared against other retrieval algorithms: for LST retrievals from Sentinel-3/SLSTR by Yang et al. (2020), where it presented the highest accuracy overall, in line with similar studies performed for other sensors. The GSW can be tailored and adapted for every region with the appropriate split-window coefficients.

Our area of interest starts at 50° latitude and encompasses the whole pan-Arctic region. The climate zones in this area differ strongly from each other, e.g., the Siberian tundra from the high mountains in Alaska or the great plains in southern Canada. The clear-sky database created by Ermida et al. (2022) is built on ERA5 data resampled with a dissimilarity criterion and includes satellite observation to determine realistic surface conditions, as opposed to the SeeBor database (Borbas et al. 2005), built from ERA-40 data. Currently most LST products (including the GLASS product) rely on the SeeBor database. The recent ERA5 exhibits significant improvement in the lower layers of the atmosphere, which improves the simulation of satellite observations performed in wavelengths more sensitive to the surface. The GSW is trained independently for each class of total column water vapor and surface temperatures. Only profiles suitable for our area of interest have been chosen in the training and testing phase.

The above points have been clarified and a more detailed description of the calibration database as well as the criteria for selecting atmospheric profiles for an LST retrieval algorithm optimized for the pan-Artic region has been included in the manuscript.

It is true that satellites have a higher coverage nearer the poles. This allows to choose scenes with viewing angles closer to nadir, which have the advantage of providing smaller footprints and higher quality data, e.g., in terms of cloud contamination and surface anisotropy. This is independent of the chosen LST algorithm. The split-window coefficients (SWC) were computed for angles up to 70°, but in the final product, all pixels with a satellite viewing angle higher than 40° were masked out to keep only the best quality data.

**2.4 Auxiliary data**

*Atmospheric profiles from the Clear-Sky Database developed at LSA-SAF (Ermida and Trigo, 2022) are used for the RT modelling (RTM). This database contains atmospheric profiles such as temperature, specific humidity and ozone on 137 model levels (full vertical resolution), sampled from ERA5 for the 2009-2019 period. The sampling technique follows the method from Chevallier et al. (2000). Surface variables like T2M, surface pressure, Tskin and emissivity are obtained from the combination of ERA5 and satellite data to ensure the best possible representation of the surface conditions. Column variables, such as TCWV and total cloud cover (TCC) are also present in the database. The atmospheric profiles are classified on TCWV varying from 0 to 60 mm and TS ranging from 190 to 340 K. The profiles belonging to our area of interest are selected.*

**3.1 Generalised Split Window algorithm**

*Based on the test sets, look-up tables (LUT) with coefficients are created for each satellite. The LUTs are organized into classes of TCWV and Tskin, allowing to allocate the right SWC to the encountered atmospheric conditions. Mean absolute error (MAE), the coefficient of determination (R2) and root mean square error (RMSE) are computed for all coefficients to keep track of the general performance of the RTM*

4. With respect to the LST validation, the authors only used KIT and SURFRAD stations. None of the stations is within the study area and therefore are not representative of the presented LST dataset. This is very clear when looking at figures 4 and 10. These stations' LSTs lowest values are around 260K, while most of the Artic is well bellow this value. There is a very with range of surface flux stations within the considered area (AmeriFlux, Fluxnet, BSRN) or even in Antarctica, which has much more similar conditions. The authors should have tried to use more stations that encompass the specifics of the polar climate. It's true that these stations tend to be more heterogeneous, but the SURFRAD stations are also very heterogeneous.

We agree that for a broader validation that is more representative of the low temperatures, high-latitude sites would be highly desirable. However, high quality in situ data from dedicated LST validation sites are rare and most of the existing stations (SURFRAD, BSRN, …), as mentioned by the reviewer, have spatial representativeness issues.

In our study we decided to only use top tier in situ LST validation data. Therefore, we only consider stations that have been investigated within the ESA GlobTemperature and the LST CCI projects in terms of their suitability for validating satellite LST and undergone quality controls. Following recommended validation protocols, in situ measurements need to have a high temporal frequency (sampling rate ranging from 1 to 3 min, according to Guillevic (2018)) to avoid additional uncertainty due to temporal mismatch / interpolation. BSRN and FluxNet only provide data averaged over a 30 min or one hour period. Furthermore, accurate emissivity information needs to be available to convert measurements of brightness temperature into in situ LST observations.

Data from the Atmospheric Radiation Measurement Climate Research Facility *US* Department of Energy (ARM) site at the North Slope of Alaska (NSA) are available from 2007 to 2012, have undergone quality control procedures and previously been used in the ESA GlobTemperature project, i.e., they meet the above stated criteria. Therefore, we integrated the in situ LST data from the NSA site into our validation and updated the corresponding figure (Fig. 4) and table (Table 2) accordingly. The surface is very heterogeneous at the NSA site, the station being close to lagoons (North Salt Lagoon and Imikpuk Lake), and very close to the coast. This explains why the performances are much worse during summertime than during winter when the entire area is snow and ice covered.

[Figure]

*Figure 4. AVHRR LST versus in situ LST at (a) Bondville (BND), (b) Desert Rock (DRA), (c) Fort Peck (FPK), (d) Goodwin Creek (GCM),(e) Penn. State Univ (PSU), (f) Sioux Falls (SFA), (g) Southern Great Plains (SGP), (h) Table Mountain (TBL), (i) Evora (EVO), (j) Lake Constance (BOD), (k) North Slope of Alaska (NSA). Red represents daytime measurements and blue represents nighttime measurements. Match-up periods are provided in the text.*

5. There is a long discussion on whether the problems in stability seen when comparing Tair with T2M and LST being related to day/night problems. It's not clear to me why the authors did not separate daytime from nighttime observations. This would make

comparing with Tair_max and Tair_min more easy to interpret. For T2M, it's not clear from the text but it seems the authors are averaging all hours of the day? The ERA5-land provides a seamless diurnal cycle with hourly frequency, why not compute the daily max and min to obtain variables comparable to Tair?

Thank you for pointing this out. We agree with the Referee: our description of the use of LST daytime data only was not clear. We now describe the time series generation more clearly. The goal was to prove the overall stability of our product based on 17 different satellites, using the ERA5-Land product. In that respect, we based our analysis on monthly mean T2M data from ERA5-Land. The highlighted topic by the Referee will be considered in our next analysis.

6. Why do you use ERA5 in some cases and MERRA-2 in other? ERA5 has better spatial and temporal resolution.

Two datasets were chosen for this study: MERRA-2 and ERA5-Land.

MERRA-2 assimilates space-based observations of aerosols and accounts for ice sheets, ensuring accurate data for regions like Greenland and Antarctica. The GLASS product (Ma, 2020) has been generated with MERRA-2 products: using MERRA-2 for our study simplifies comparisons between the GLASS and our product. Our study uses skin temperature and total column water vapor from MERRA-2 to determine the atmospheric conditions at each pixel and to select the SW coefficients from the look up table.

In contrast, air temperature data from ERA5-Land are used for the correlation and stability analysis. Using separate sources of reanalysis data helps keeping the stability analysis independent from the SW coefficient assignment process. Furthermore, ERA5-Land has been fully validated in the Arctic region in a previous study on trend analysis (Rantanen, 2023, ARCLIM atlas). Finally, in the stability analysis we compared point data (weather station data), satellite data and a reanalysis product: in order to optimize the spatial representativeness in the point-to-pixel comparisons, the reanalysis product with the highest spatial resolution has been chosen.

The above choices are now explained in a corresponding paragraph added to subsection *#2.4 Auxiliary data*.

**2.4 Auxiliary data**

*Skin temperature (Tskin) and Total Column Water Vapor (TCWV) from the MERRA-2 reanalysis dataset (M2T1NXSLV, variables are labelled TS and TQV). The data come at hourly temporal resolution with a spatial resolution of 0.5° x 0.625°. Nearest neighbour resampling was performed to match the AVHRR spatial resolution and scanline time, i.e. as in the work*

*of Ma et al. (2020). MERRA-2 is preferred over other reanalysis products with finer spatial resolution to allow comparison with the GLASS product (Ma et al. 2020) and to keep the LST retrieval independent from ERA5-Land, which will be used for the stability analysis.*

7. Is/will this dataset be made available publicly? What is the format? What is the projection? More technical details about the dataset are needed.

This product is part of a PhD project, and part of ongoing research, but we consider releasing it after completion of the project.

The PyGAC AVHRR FDR from EUMETSAT (2023) is available in the Network Common Data Form (NetCDF) format and so is our Pan-Arctic LST product. We kept the same data structure as the original FDR. The dataset covers the pan-Arctic region (− 180°, 90°, 180°, 50°) at a spatial resolution of 0.05 × 0.05° pixel size. Technical details regarding format and projection are added to the manuscript as well as details on the format of the EUMETSAT dataset.

**2.1 EUMETSAT AVHRR FDR**

*The IR calibration procedure is satellite-specific, with no cross-calibration between satellites for IR channels (EUMETSAT, 2023d). The PyGAC AVHRR FDR from EUMETSAT (2023) is available in the Network Common Data Form (NetCDF) format and covers the entire globe (− 180°, 90°, 180°, -90°) at a spatial resolution of 0.05 × 0.05° pixel size. This study focuses on the pan-Arctic region, therefore only data above 50° N have been processed.*

**References**

Borbas, E.E.; Seemann, S.W.; Huang, H.L.; Li, J.; Menzel, W.P. Global profile training database for satellite regression retrievals with estimates of skin temperature and emissivity. In Proceedings of the International TOVS Study Conference-XIV, Beijing, China, 25–31 May 2005.

EUMETSAT (2023): AVHRR Fundamental Data Record - Release 1 - Multimission, European Organisation for the Exploitation of Meteorological Satellites, DOI: 10.15770/EUM_SEC_CLM_0060. http://doi.org/10.15770/EUM_SEC_CLM_0060

Guillevic, P., Göttsche, F., Nickeson, J., Hulley, G., Ghent, D., Yu, Y., Trigo, I., Hook, S., Sobrino, J.A., Remedios, J., Román, M. & Camacho, F. (2018). Land Surface Temperature Product Validation Best Practice Protocol. Version 1.1. In P. Guillevic, F. Göttsche, J. Nickeson & M. Román (Eds.), Good Practices for Satellite-Derived Land Product Validation (p. 58): Land Product Validation Subgroup (WGCV/CEOS), doi:10.5067/doc/ceoswgcv/lpv/lst.001

Karlsson, K.-G., Stengel, M., Meirink, J. F., Riihelä, A., Trentmann, J., Akkermans, T., Stein, D., Devasthale, A., Eliasson, S., Johansson, E., Håkansson, N., Solodovnik, I., Benas, N., Clerbaux, N., Selbach, N., Schröder, M., and Hollmann, R.: CLARA-A3: The third edition of the AVHRR-based CM SAF climate data record on clouds, radiation and surface albedo covering the period 1979 to 2023, Earth Syst. Sci. Data, 15, 4901–4926, https://doi.org/10.5194/essd-15-4901-2023, 2023.

Ma, J., Zhou, J., Göttsche, F.-M., Liang, S., Wang, S., and Li, M.: A global long-term (1981–2000) land surface temperature product for NOAA AVHRR, Earth Syst. Sci. Data, 12, 3247–3268, https://doi.org/10.5194/essd-12-3247-2020, 2020

Pedelty J., Devadiga S., Masuoka E., Brown M., Pinzon J., Tucker C., *et al*.
Generating a long-term land data record from the AVHRR and MODIS Instruments. Geoscience and Remote Sensing Symposium, 2007. IGARSS 2007, Barcelona (2007), pp. 1021-1025

Solberg, R., G. Schwaizer, T. Nagler, S. Wunderle, K. Naegeli, K. Luojus, M. Takala, J. Pulliainen, J. Lemmetyinen, and M.
Moisander (2021) ESA CCI+ Snow ECV: Product User Guide, version 3.1, December 2021.

Wan, Z. and Dozier, J.: A generalized split-window algorithm for retrieving land-surface temperature from space, IEEE T. Geosci. Remote, 34, 892–905, https://doi.org/10.1109/36.508406, 1996.

Yang, J., Zhou, J., Göttsche, F. M., Long, Z., Ma, J., and Luo, R.: Investigation and validation of algorithms for estimating land surface temperature from Sentinel-3 SLSTR data, Int. J. Appl. Earth Obs., 91, 102 136, https://doi.org/10.1016/j.jag.2020.102136, 2020.

Zhou, J., S. Liang, J. Cheng, Y. Wang, and J. Ma, 2019: The GLASS land surface temperature product. *IEEE J. Sel. Top. Appl. Earth Obs. Remote Sens.*, **12**, 493–507, https://doi.org/10.1109/JSTARS.2018.2870130.

---

## Author Response (AR2)

Author's response

We thank the Reviewers and Editor for their respective feedback.

Point-by-point Answers

We thank the Reviewer for the useful and detailed comments. We followed the suggestions and modified our manuscript as described in the line-by-line answers below. Note that the reviewer's comments are in black, our answers are in red, and the changes made to the manuscript are in red italic.

**The authors have addressed some of my questions, which were mostly related to some details not being clear in the text. I still think there are limitations in the in-situ validation and that more stations with similar climate should be added to the paper. I would even remove stations at lower latitudes. See below point-by-point reply.**

**1. I understood that the authors select the satellites in order to reduce time differences due to orbital drift and that there is no overlap between satellites. But how do you handle orbit overlap from the same satellite? At the poles there is significant overlap of the orbits resulting in multiple observations throughout the day. Are these all averaged? Do you select a specific orbit depending on the time?**

For each day and pixel, for each satellite, the dataset contains values selected from two single orbit files (one for daytime and one for nighttime), i.e., no averaging is performed. The selected observation is the one closest to the zenith (NADIR condition). This ensures that the observations are made at the same viewing conditions and at nearly the same local time at a hemispheric scale. The subsection *2.1 EUMETSAT AVHRR FDR* now clarifies this point.

*2.1 EUMETSAT AVHRR FDR*

*For each satellite, the orbit files are composited by choosing for each pixel only the observation closest to nadir. The composited files have a spatial resolution of 0.05° x 0.05° pixel size and are available for each satellite twice a day (at daytime and nighttime). This study focuses on the pan-Arctic region, therefore only data above 50° N have been processed.*

**3. If the calibration is limited to your area of interest, then why use stations outside such area to validate the LST?**

The radiative transfer modelling is done independently for each class of TCWV and Tskin, and the corresponding split-window coefficients (SWC) are then obtained for each of these classes in an independent manner. Therefore, for each pixel, the calibration is performed for the adequate class of TCWV and Tskin, i.e., it is not based on an area of interest. The Pan-Arctic pixels are calibrated with classes corresponding to actually encountered atmospheric

conditions; the same logic applies to pixels over the USA, which are then also validated with SURFRAD stations. We modified our description of the calibration process (*3.1 Generalized Split Window Algorithm*) to clarify this point and remove the sentence that is confusing in *2.4 Auxiliary data.*

*3.1 Generalized Split Window Algorithm*

*Table 3 summarises the construction of the simulation dataset. Finally, the calibration was performed independently for each class, and for each class, the samples were split into a training (70%) and test (30%) set, and multilinear regression was performed on the resulting BTs. Based on the test sets, look-up tables (LUT) with coefficients are created for each satellite.*

**4. Again, the authors agree that the SURFRAD stations have large representativeness issues, so I do not see how they can be "top-tier". Also, a lot of BSRN stations have 1 minute sampling so I don't understand why the authors don't want to use them. There are also a few high latitude stations available from the Copernicus Ground-Based Observations for Validation (GBOV) service. Regarding the accuracy of emissivity, I believe this is true for all stations and, to my knowledge, none of the stations used by the authors have in-situ measured emissivity. There are specific limitations of the retrievals close to the poles than can only be properly assessed with stations in these areas. For instance, cloud contamination tends to be much high for night-time snow covered surfaces and, as such, selecting stations with these conditions could help clarify how much cloud contamination could be in the data.**

The SURFRAD stations are regarded as 'top-tier' mainly because of their long and quality-controlled time series. Furthermore, the surface around them (land cover types, seasonality, etc.) has been studied in detail. A lack of spatial representativeness is something that SURFRAD stations share with BSRN stations: this can be partially overcome by performing validations only at night-time. However, there is a lack of high-quality in-situ data obtained in large, spatially homogenous areas. GBOV mainly collects data from existing stations (SURFRAD, BSRN, e.g.). Therefore, GBOV data have the same issues, e.g., length of time series, quality, suitability (sensors) and representativeness. The emissivity at densely vegetated sites or complete snow-covered sites is close to one and can be estimated very accurately.

We have added additional station data from the Baseline Surface Radiation Network (BSRN) located in the northern high latitudes. Among the few stations available at high latitudes, we selected three stations:

ALE (Alert, Lincoln Sea, Canada)

NYA (Ny-Ålesund, Spitsbergen)

TIK (Tiksi, Siberia, Russia)

The remaining arctic stations were not considered as either not enough data points were available or the downward radiation components were missing. We show a comparison of satellite-based and in-situ LST data in Figure 5 (plot visible below). In-situ LST for the BSRN stations is computed from their radiation components as in:

Martin, M.A.; Ghent, D.; Pires, A.C.; Göttsche, F.-M.; Cermak, J.; Remedios, J.J. Comprehensive In Situ Validation of Five Satellite Land Surface Temperature Data Sets over Multiple Stations and Years. *Remote Sens.* **2019**, *11*, 479. https://doi.org/10.3390/rs11050479.

Specifically, broadband emissivity (BBE) is obtained from channel effective emissivity data provided in the ASTER GED with the linear equation described in:

J. Cheng, S. Liang, Y. Yao and X. Zhang, "Estimating the Optimal Broadband Emissivity Spectral Range for Calculating Surface Longwave Net Radiation," in *IEEE Geoscience and Remote Sensing Letters*, vol. 10, no. 2, pp. 401-405, March 2013, doi: 10.1109/LGRS.2012.2206367.

The terrain corresponding to one GAC pixel around NYA is very heterogeneous (mountainous terrain), leading to poorer validation results than the two other BSRN stations. The BSRN stations are also all located close to the shoreline and snow-covered part of the year.

Due to the fact that only cloud-free data sets are used for the matchup with in-situ stations, the different cloud coverage in the Arctic modifies the number of selected matchups but not the quality of the comparison.

Section *2.4 Auxiliary data* and Table 2 (description of used stations and procedure to derive LST) has been changed accordingly. The Figure 5 has been added to the Section *4.1.2 Validation with in situ LST*

[Figure]

*Figure 5 : AVHRR LST versus in situ LST at BSRN stations (a) Alert, Lincoln Sea (ALE), (b) Ny-Ålesund, Svalbard (NYA) and (c) Tiksi, Russia (TIK). Red represents the daytime measurements, and blue represents the nighttime measurements. Match-up periods are provided in the text.*